# Rat anterior cingulate cortex recalls features of remote reward locations after disfavoured reinforcements

**Ali Mashhoori†, Saeedeh Hashemnia†, Bruce L McNaughton, David R Euston, Aaron J Gruber\***

Canadian Centre for Behavioural Neuroscience, Department of Neuroscience, University of Lethbridge, Alberta, Canada

**Abstract** The anterior cingulate cortex (ACC) encodes information supporting mnemonic and cognitive processes. We show here that a rat's position can be decoded with high spatiotemporal resolution from ACC activity. ACC neurons encoded the current state of the animal and task, except for brief excursions that sometimes occurred at target feeders. During excursions, the decoded position became more similar to a remote target feeder than the rat's physical position. Excursions recruited activation of neurons encoding choice and reward, and the likelihood of excursions at a feeder was inversely correlated with feeder preference. These data suggest that the excursion phenomenon was related to evaluating real or fictive choice outcomes, particularly after disfavoured reinforcements. We propose that the multiplexing of position with choice-related information forms a mental model isomorphic with the task space, which can be mentally navigated via excursions to recall multimodal information about the utility of remote locations.

**\*For correspondence:**
aaron.gruber@uleth.ca

†These authors contributed equally to this work

**Competing interests:** The authors declare that no competing interests exist.

## Introduction

The ACC and other nearby structures in the medial prefrontal cortex (mPFC) play an important role in the control of both memories and decisions (*Euston et al., 2012*). These structures influence memory retrieval via connectivity with the hippocampus (*Ito et al., 2015*; *Rajasethupathy et al., 2015*), and are thought to utilize hippocampal output to form semantic or schematic knowledge of the world from past experience (*McClelland et al., 1995*). Activation of patterned neural activity in the mPFC may thus play an important role in utilizing experiential or schematic knowledge to plan or control behaviour (*Tse et al., 2007*; *Wang et al., 2012*). Neurons in the ACC and nearby regions encode a variety of task features related to reinforcement and decisions (*Kennerley et al., 2006*; *Gruber et al., 2010*; *Sul et al., 2011*), and many are also selectively active over large regions of the task space (*Jung et al., 1998*; *Euston and McNaughton, 2006*; *Fujisawa et al., 2008*; *Jadhav et al., 2016*). The function of this broad and distributed spatial mapping by individual ACC units has remained more contentious than the sparse encoding of location by neurons in the hippocampus (*Burton et al., 2009*; *Hyman et al., 2012*).

The hippocampus contains 'place cells' that provide information about the position of an animal in an environment (*O'Keefe and Dostrovsky, 1971*; *Wilson and McNaughton, 1993*), which is utilized by the ACC (*Burton et al., 2009*). Although ACC neurons had generally been thought not to generate place fields (*Poucet, 1997*), recent work has revealed neurons in mPFC with spatially specific firing that typically span over 50 cm, and that are distributed over the task environment (*Fujisawa et al., 2008*). These are properties expected of a place code. Nonetheless, the broad spatial encoding in the mPFC has often been interpreted as signaling contextual features, such as the environment (*Hyman et al., 2012*) and task (*Ma et al., 2016*). The ACC also appears to utilize reinforcement information from past actions to engage action strategies that improve cost-benefit

outcomes (*Botvinick et al., 2004*; *Amiez et al., 2006*; *Kennerley et al., 2006*; *Rushworth et al., 2011*; *Heilbronner and Hayden, 2016*), particularly when rapid shifts in strategy are needed to optimize reward acquisition (*Posner et al., 2007*; *Passingham and Wise, 2012*). One of its specific functions is to encode unattained rewards (*Hayden et al., 2009a*), which may contribute to its role in signalling regret when outcomes do not meet expectations (*Coricelli et al., 2005*). It is possible that the ACC uses a fine-grained spatial map as a mnemonic scheme to recall possible alternate outcomes at other locations for such processing, but we are unaware of any direct evidence with sufficient spatiotemporal resolution to accurately decode such shifts.

## Results

### Precise spatio-temporal encoding of position by ACC

Because populations of broadly tuned cells can encode quantities with more precision than the encoding of individual cells (*Kim et al., 2012*), we first sought to determine whether population activity in the ACC accurately encodes the position of an animal. We recorded ensembles of ACC neurons while rats performed a binary choice task on a figure-8 track (*Figure 1A*). Two target feeders at the north corners of the track could be reached after turning right or left at the choice point, and a third feeder in the central segment was used to motivate rats to return to the starting position. Rats received reinforcement at the central feeder on every trial. The effort-reward utility of each choice was independently controlled by elevating the target feeders to one of three heights and by providing either a small or large reward volume. The utilities were held fixed for 16 consecutive trials and the animals were forced to alternate between the right and left options on the first 10 trials before being allowed free selection of either option for the remaining 6 trials of the block. The task was run in a 6-block sequence of 96 trials (*Figure 1B*), and the same sequence would restart

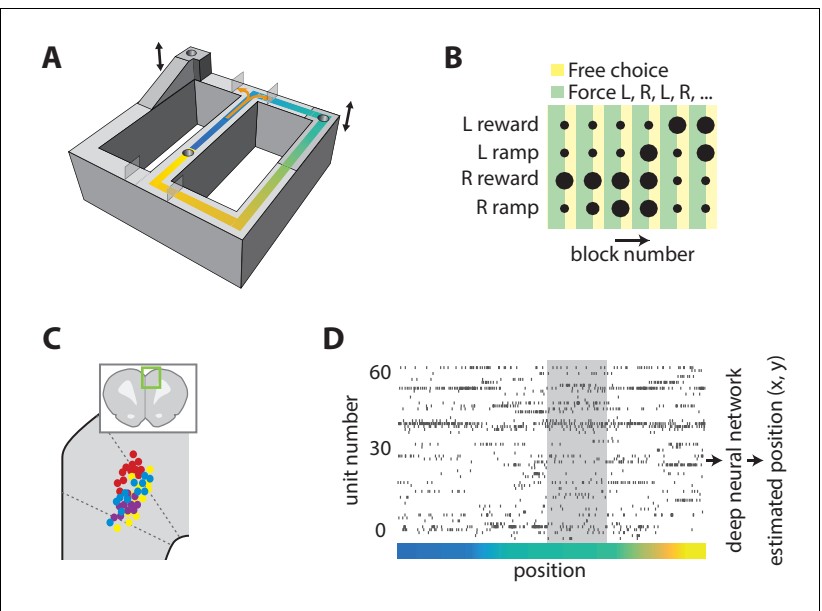

**Figure 1.** Task and neural recording. (A) Schematic illustration of the figure-8 track, showing the locations of the feeders (cylindrical depressions), height-adjustable platforms (indicated by arrows), and movable gates (translucent rectangles). (B) Graphical representation of the choice reward-effort utilities (dot size) and choice option (color) structure of one task session. The effort-reward utility of each choice was constant during each block of 16 laps. (C) Illustration of estimated recording locations in the dorsal medial prefrontal cortex (inset), showing that most fell in the ACC. (D) Representative example of simultaneously recorded ACC ensemble activity during one lap of the task. The color indicates the position on the track as coded in panel A, and the grey shaded region is corresponds to the target feeder location.

The online version of this article includes the following figure supplement(s) for figure 1:

**Figure supplement 1.** Spatial attributes of ACC unit firing and decoding error.

upon completion within each session. The reward contingencies were counterbalanced among right and left feeders between sessions. Rats completed 162–256 trials per session. Animals received the same small reward at the central feeder. We simultaneously recorded position and neural activity in the ACC (*Figure 1C–D*), and found that most cells activated over large areas of the track (*Figure 1— figure supplement 1*), consistent with previous reports.

The currently predominant neural decoding model for position is Bayesian reconstruction (*Brown et al., 1998*; *Zhang et al., 1998*; *Carr et al., 2011*). We found, however, that we could achieve significantly lower decoding error (36% reduction; t(6) = 9.0, p=0.0001, power = 1) than the Bayesian method by using a deep artificial neural network (dANN) to decode location from patterns of neural activity in bins of 20–50 ms (*Figure 2B*; see Materials and methods). The dANN was memoryless in that it only used information from the present time bin for the predictions. It could exploit higher-order statistical relationships among inputs than could the Bayesian method, and could learn to ignore spurious information. Its superior performance therefore likely demonstrates that either these higher-order statistics carry a significant amount of information about spatial position, as previously predicted (*Fujisawa et al., 2008*), or that the representation of non-spatial features hampers Bayesian reconstruction. Our analysis of multiple sessions from four animals revealed that approximately 30 randomly selected ACC units are required for good reconstruction accuracy, whereas near asymptotic accuracy can be achieved by using the 17 most informative cells (*Figure 1—figure supplement 1*). We therefore focused subsequent analysis on seven sessions from two animals with at least 40 simultaneously recorded cells so as to achieve decoding error close to the apparent asymptotic limit.

We next sought to determine the robustness of the decoder by quantifying its performance in the presence of noise. We shifted each spike time by a randomly drawn value from a distribution that was proportional to the variance of the cell's firing interval (25% of its STD). The dANN was trained on some trials of the uncorrupted data, and tested on different trials with the noisy data (mean spike shift was 25 ms). This increased the error (RMSE) by 1.3% (0.14 cm), which was a small yet statistically significant difference (t(6) = 4.8; p=0.003; power = 0.95). This indicates that spike timing on the order of 25 ms contained useful information for the decoding with the present neuronal sample size. Furthermore, the noise did not cause any large deviations in decoding. Only 0.07% of samples deviated more than 25 cm from the original decoding, and none were more than 50 cm (*Figure 2C*). These data indicate that the decoder is robust against moderate levels of spike jitter. The median decoding accuracy of the rat's position on the task was less than 10 cm (*Figure 2D–E*). This is much less than the length of the animal's body, and much less than the spatial selectivity of individual cells (*Fujisawa et al., 2008*).

## Invariance of spatial encoding during task sessions

We analyzed the stability of the spatial information over the entire maze through time. For this, we separated all the trials that belonged to the trial configuration (reward, effort, direction) with the maximum number of trials in each session. Then, we performed two different tests. In one, we created the training set by selecting every other trial in the list and used the remaining trials as the test set. In the second test, we used the first half of the trials as the training set and used the second half of the trials as the test set. If the spatial encoding shifted as a result of the intervening configurations (block types), or drifted in time, then the decoding accuracy in these two cases should be different. We found that decoding errors were not significantly different in these two cases (0.6 cm change in error, $t_6$ = 1.28, p=0.24). This analysis provides some evidence that the position-related features of neural activity used by the decoding network are stable within the session, despite changes in effort and reward contingencies that occur during intervening blocks.

## Excursions of spatial encoding from the physical position to a feeder

The mean spatial decoding error was low in all positions on the track (*Figure 2F–H*). The decoded position, however, sometimes deviated from the actual position by up to nearly 100 cm, and these large excursions occurred almost exclusively at the two choice-option feeders (*Figure 2G–H*). These excursions were not random in time or decoded position. Rather, the excursions consisted of several consecutive points encoding the alternate target feeder before returning to the present location of the rat (*Figure 3A–B*; *Video 1*). The localization of these excursion endpoints was particularly

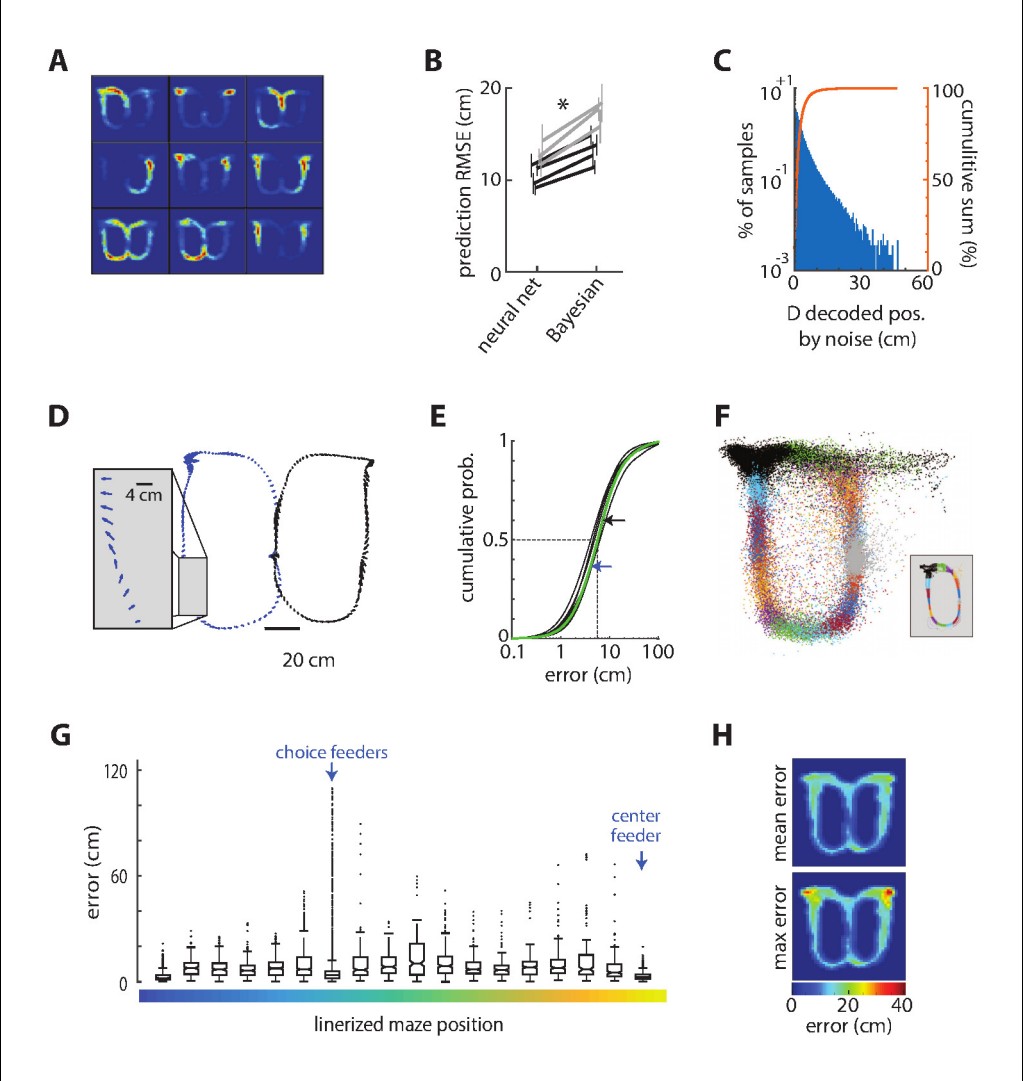

**Figure 2.** Decoding position from ensemble ACC activity. (A) The spatial selectivity of the nine most informative ACC cells for decoding position in one session chosen by the decoding network. Spike density on the track is coded by color, from blue to red. The units are rank ordered by importance from top left to bottom right. (B) The root mean squared error (RMSE) of the position decoded from the ACC activity as compared to the actual position of the rat for each session (line) and rat (shade), showing that the deep artificial neural network generates lower prediction error than does a Bayesian decoder for each of the seven sessions tested. These session-averaged errors are inflated by occasional large errors around the reward zones, as described below. Error bars show standard deviation of 20 randomly selected training and test sets for each session and method. (C) The distribution of changes in decoded position by noise. (D) Error vectors for two representative laps of the task. The arrows indicate the magnitude and direction of the decoding error every 50 ms. (E) Cumulative probabilities of the prediction error magnitude for the seven sessions. The dotted lines indicate the median, and the arrows indicate the median error for the left (blue) and right (black) laps session shown in panel C (green curve). (F) Decoded position for test data from one session, color coded by the actual position (inset) (G) The error computed every 50 ms in one representative session, represented as a box plot according to track position as shown in *Figure 1*. The box plot shows the median (horizontal lines in boxes), 95% confidence intervals (notches), first and third quartiles (boxes ends), and outliers (dots). There are a disproportionate number of outliers in the bin corresponding to the target feeder locations, but the median prediction accuracy is as good at these feeders as anywhere else on the track. (H) The mean (top) and maximal (bottom) prediction error for discretized positions on the track, showing that the very large errors occurred exclusively at the location of the target feeders. These show mean of means computed from all sessions.

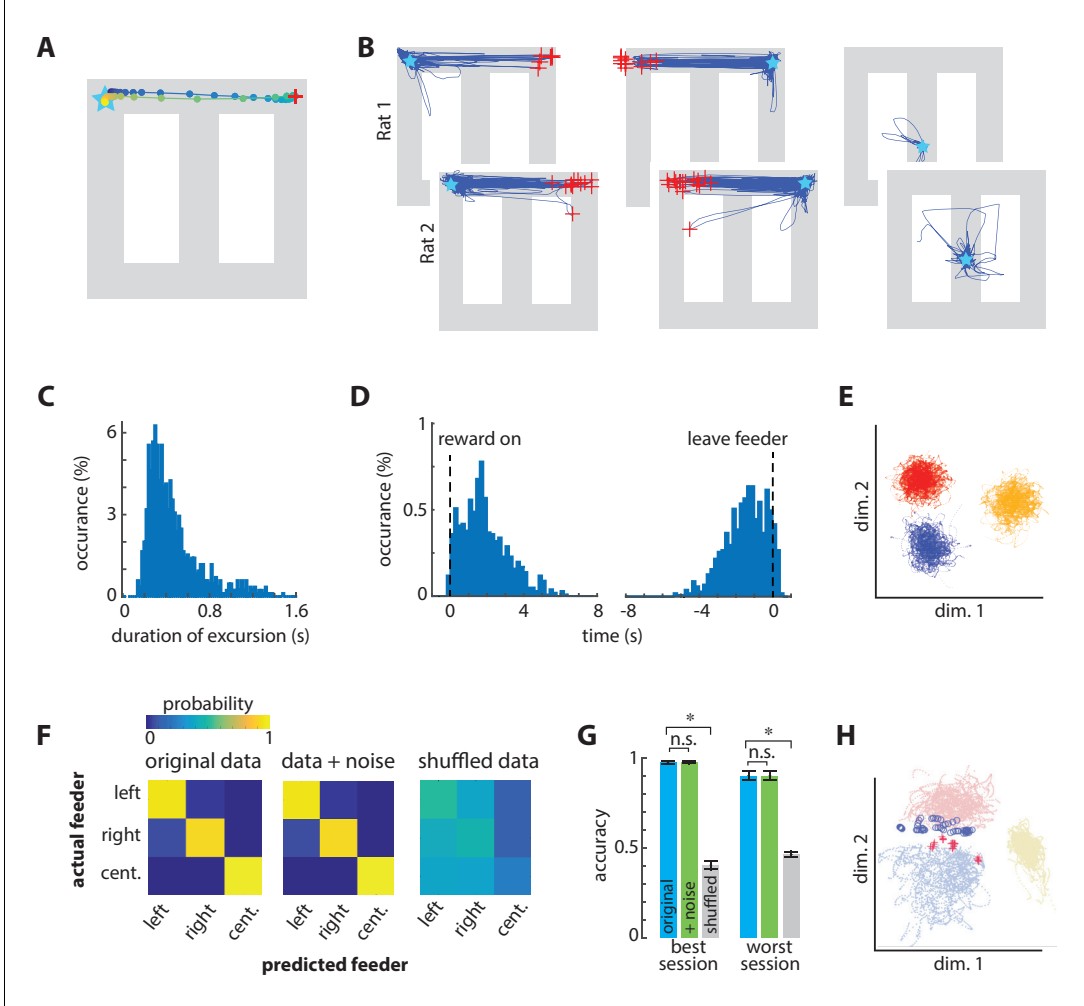

**Figure 3.** Excursions of encoded position sometimes shift to the alternate choice feeder. (A) Example of one excursion episode. The decoded position is indicated by circles plotted every 20 ms. The excursion sweeps from the actual position of the rat (star) to the right-side feeder, and then returns. The red '+' indicates the maximal prediction error. (B) All excursions from the three feeders in the test set from one session in each rat. The red '+' indicate the maximum error distance from the occupied feeder of at least 70 cm. The excursions from the right/left target feeders generate a trajectory to the alternate target feeder location. (C) Frequency distribution of excursion duration from all sessions. (D) Timing of excursion onset aligned to feeder activation or feeder zone exit, showing that excursions occur predominantly between these events. (E) Neural activity is distinct at the three feeder zones, as shown here by linear discriminant analysis of the smoothed neural data. Dots are the smoothed and binned neural patterns at feeders in the absence of excursions, and each color indicates data from one feeder (center feeder is yellow). (F) Confusion matrices for identifying the three feeders based on neural activity. Shifting each spike time by a random value (indicated as +noise) had little effect on the ability to correctly identify the feeders, whereas fully shuffling the inter spike intervals eliminated discriminability. (G) The prediction accuracy of classifying feeders for the best and worst sessions. Data in panels F and G include excursion events, which degrade performance. Asterisks indicate significantly different means with a p<0.001. (H) PCA of activation in the middle layer of the spatial decoding network for non-excursion (dots) and excursion (+and o) patterns at target feeders (blue, red) and center feeder (yellow).

striking because our decoder network output only an x and y coordinate, with no constraints that the decoded position lie on the track. This pattern of endpoints is therefore exceedingly unlikely by chance ($\chi^2$ = 484, p=1 E-40, power = 1). These excursions did not originate or terminate at the center feeder, even though the animal received the same reward type at all feeders, and the volume at the center feeder was equivalent to the small volume at the choice feeders. These features strongly suggest that the excursion phenomenon is involved in computations related to evaluating the choice options, and not related to general qualities of the reward or to planning the immediately next action, which was always a return to the center feeder as enforced by gates on the track.

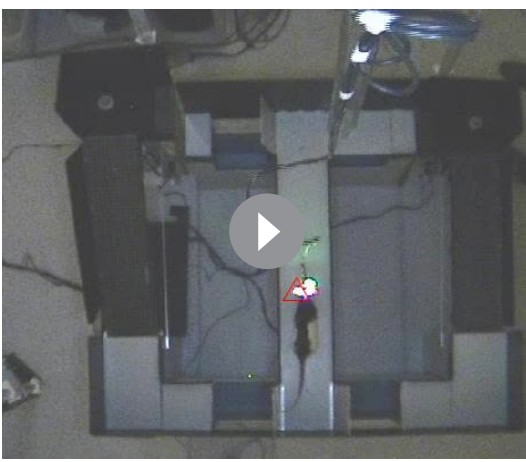

**Video 1.** Montage showing laps of the task in which the Excursions events occur at the right-hand feeder. The position decoded from neural activity is indicated by the centroid of the red triangle, which is superimposed on top of the overhead video of the rat's actual position on the track. The head-mounted LED lights are used as the target position. The decoded position tracks the LEDs reliably, but generates an excursion to the left-hand feeder after reward consumption.
https://elifesciences.org/articles/29793#video1

The median duration of the excursion events was 400 ms (20 consecutive bins of 20 ms; *Figure 3C*), and occurred almost exclusively while the animal was stationary at the feeders. Although excursions occasionally occurred during the reinforcement, most occurred after the reinforcement and prior to locomotion away from the feeder (*Figure 3D*). The duration of the excursions was appreciably longer than the width of the smoothing kernel (120 ms), so the end-points of the excursions were not affected by the smoothing. The dynamics of the transition, however, occurred on a time scale less than the kernel width and was therefore strongly affected. The intermediate points of excursion between the feeder sites appear to be a blend of the encoding of the two feeder sites coming from the neural dynamics and/or the smoothing kernel in our analysis. We therefore suggest that the phenomenon is more accurately conceptualized as a shift rather than a replay of the true trajectory in the physical space.

We next sought further independent evidence as to whether the excursion events were an artifact of misclassification rather than a neurophysiological phenomenon. One possibility is that both target feeders are encoded by similar ensemble activity because of shared reward encoding, or some other feature, so that the network might confuse left and right feeders such that the decoded position might appear to jump from one to the other. To address this concern, we first used linear discriminant analysis to test if the neural patterns at the feeders are distinct from one another. Linear discriminant analysis of the input activity patterns in the absence of excursion did form distinct clusters for each feeder, indicative of unique pattern features at each feeder (*Figure 3E*). This method is independent of our decoder, and therefore provides graphical validation. In order to quantify the pattern separation, we trained a new 3-layer network to classify patterns of neural activity from each of the three feeders, and then tested the classification accuracy (via cross-validation) on either samples of the original data, original data corrupted by noise, or fully shuffled data (*Figure 3F–G*). These tests include all trials, both with and without excursion events. If the excursion events were due to a classification error arising from similarity of neural patterns at the feeders, then adding noise should decrease accuracy. The noise did not significantly reduce feeder decoding accuracy (t(6) = 1.02; p=0.35, power = 0.87), but fully shuffling the spike time intervals did (t(6) = 32.6; p=3e-8, power = 1). These data indicate that the activity patterns at the feeders are sufficiently distinct such that the introduction of noise does not cause misclassification, suggesting that the excursions are not a consequence of small random variation of the inputs.

The analyses above do not rule out the possibility that the excursions arise from brief shifts from the unique activity features at each feeder to an activity state common to both feeders. For instance, the reward encoding neurons could strongly activate to overshadow the position information in some instances, and this could produce a pattern that emerges at both feeders, but is distinct from the normal encoding at the feeders, and thereby confuses the decoder. This should be apparent in the variance of patterns at target feeders represented by the decoder network. PCA of activation in layer 3 of the position decoder shows distinct clusters for the non-excursion patterns at the target feeders (*Figure 3H*). Moreover, the excursion patterns do not overlap completely or form their own cluster, and instead tend to overlap with the unoccupied target feeder cluster. We next conducted an independent and quantitative test for a common state by computing the classification accuracy of untransformed ACC patterns (input to the network) among the four classes: feeder A during an excursion (A'); feeder B during an excursion (B'); feeder A not during excursion (A); and feeder B not

during excursion (B). If the excursions are because of a transition to a common state from both feeders, then the excursion patterns should be highly discriminable from the non-excursion patterns at the same feeder (A' from A, and B' from B), but not discriminable from each other (A' from B'). We found strong evidence for the former, but not the latter. We used the area under the curve (AUC) of the receiver operator characteristic (ROC) to quantify discriminability of samples from pairs of these conditions. An AUC value close to 0.5 indicates that the patterns from two classes are indiscriminable, whereas an AUC value close to one indicates that patterns are highly discriminable by the classifier. AUC values between these limits indicate that features of the patterns are sometimes similar and sometimes dissimilar in at least some dimensions. The discrimination of excursions from non-excursions at the same feeder (A' from A, and B' from B) was very high (AUC = 0.94). On the other hand, the patterns during the excursions from the two feeders (A' from B') were discriminable at a moderate level (mean AUC = 0.75). This latter analysis was limited in power because of the low number of excursions at the preferred feeder relative to the dimensionality of the patterns. Nonetheless, these data indicate that the untransformed input patterns during excursions can often be distinguished based on the position of the rat. The non-linear transform of the input by the middle layers of the decoding network apparently separates the excursions from one another, but not from the typical feeder patterns (*Figure 3H*). In sum, the non-excursion patterns (A, B) are highly discriminable (e.g. *Figure 3E–G*), as are the excursion patterns from the non-excursion patterns at the target feeder (A, A', and B, B'). The excursion patterns sometimes overlap with each other (A', B'), and with the pattern from the unselected feeder (A', B and B', A; *Figure 3B,H*). It thus appears that some features of the encoding shift to be more similar to the unselected feeder during excursion events. Because reward and location were confounded in the experimental design, we cannot rule out the possibility that reward encoding contributes to the phenomena. The inability to fully discriminate the excursion patterns from one another could involve some feature of the reward, such as volume, which flips between the choice feeders during the session. We therefore next investigated if units encoding reward were activated during the excursions.

## Excursions were more likely to occur at less-preferred feeders, and encoded reward and choice information

Rats developed a strong feeder preference during free-choice trials within each block because of the unequal effort-reward utilities. Within each block of 6 free-choice trials, the preferred feeder was chosen an average of $5.3 \pm 0.5$ times. We used this choice bias as a measure of revealed preference among feeders, and analyzed the 10 forced-choice trials that preceded it. The selective occurrence of the excursions at the right-side feeder site in these forced-choice trials was strongly anti-correlated with the revealed preference for this feeder on free-choice trials (*Figure 4A*; $r^2 = 0.86$; $F(6) = 31.5$; $p=0.002$; power = 0.97). In other words, the excursion was much more likely to occur when the rat was forced to select the less-preferred feeder. The excursions also emerged more frequently in free-choice trials when rats chose the less-preferred option. This dependence suggests that excursions are related to disfavored outcomes, consistent with proposals that primate ACC is involved in regret or signalling other outcomes that could have occurred (*Coricelli et al., 2005*; *Hayden et al., 2009a*). This supposition predicts that excursions should contain information related to the value of choice options. We thus next sought to determine if cells encoding choice or reward become activated during excursions. We first used logistic regression with norm-1 regularization to determine the degree to which cells encoded reward or choice information at the feeder locations in the absence of excursions (see Materials and methods). We found that 30% and 36% of cells were strongly predictive of reward and choice, respectively. We next independently determined which cells in the population significantly increased firing during excursions. This analysis revealed that some cells predictive or reward and/or choice activated during the excursions (*Figure 4B*).

## Discussion

We have shown here that the head position of a rat on a track can be decoded from the activity of several dozen ACC neurons with an accuracy of about 10 cm. This is a much finer spatial scale than the very broad spatial sensitivity of individual neurons in the ACC and nearby mPFC (*Fujisawa et al., 2008*). This raises the possibility that ACC ensembles may represent environmental features on a fine spatial resolution. Moreover, we found that the encoded position normally tracked the current

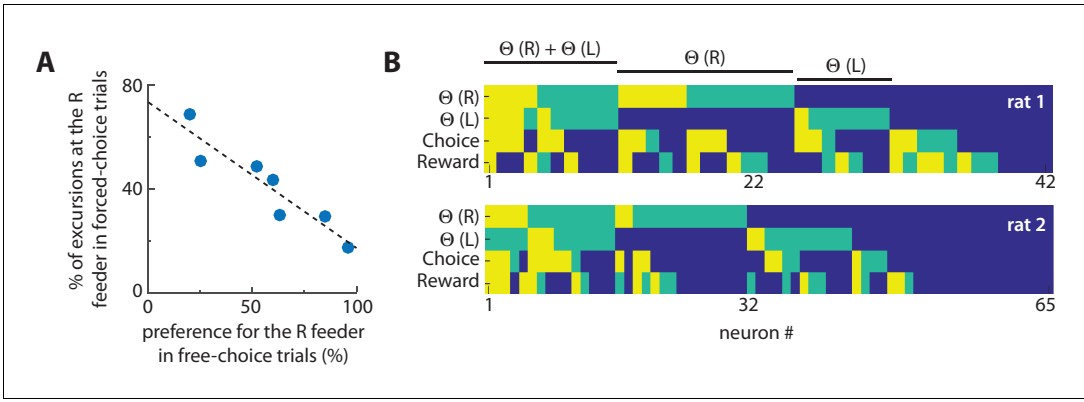

**Figure 4.** Excursions are more likely at non-preferred feeders, and encode choice-reward information. (**A**) The relationship between excursions frequency and feeder preference. Each dot is the session-averaged occurrence of excursion events at the right-hand (R) target feeder (as % of all events) during forced-selection trials, plotted against the revealed preference for that same feeder computed by the choice bias to the right-hand feeder in free selection trials (% of all choices). The negative correlation reveals that the excursion phenomenon is more likely to occur at disfavored feeders. (**B**) Encoding of information related to excursion, choice, and reward among neurons. The relative information of each neuron for discriminating excursions at the left target feeder (Θ(L)), excursions at the right target feeder (Θ(R)), the feeder choice, and the reward level. The level of information was determined by the frequency it was used by a neural network to discriminate this information, and is categorized as very informative (yellow), somewhat informative (green), or uninformative (blue). This analysis shows that some cells involved in excursions also encode information about choice and/or reward.

state of the rat, but sometimes dissociated from its physical position at the target feeders. These excursions did not occur at the central feeder, were more likely to occur at the disfavoured target feeder, and involved the activation of neurons encoding reward and choice. We suggest that this is functional evidence for the evaluation of choice outcomes that is more likely to occur following disappointing reinforcements. Moreover, the preponderance of evidence suggests that the ACC encoding during most excursions became more similar to the unselected target feeder than the selected one. If so, the data provide evidence that the ACC evaluates unrealized choice outcomes at locations remote from the animal's position.

These data are consistent with many previous findings and proposals. First, indirect evidence has previously suggested that ACC and adjacent areas in mPFC encode the position of the animal and objects on a fine spatial scale. For instance, small deviations in running path has been shown to explain a significant amount of variance in rat ACC activity (*Euston and McNaughton, 2006*; *Cowen et al., 2012*), and mPFC lesions impair object-in-place memory but not object memory (*Barker et al., 2007*). Second, ACC encodes a variety of task-related information such as reward, choice, and effort, which is thought to support choices among options with differing costs and benefits (*Botvinick et al., 2004*; *Amiez et al., 2006*; *Euston and McNaughton, 2006*; *Kennerley et al., 2006*; *Euston et al., 2007*; *Rushworth et al., 2011*; *Cowen et al., 2012*; *Heilbronner and Hayden, 2016*). A specific function of individual ACC units in monkeys is the signalling of fictive outcomes, which are potential reinforcements that did not occur (*Hayden et al., 2009a*). This could be analogous to the activation of reward-encoding units during excursions, although the difference in species, spatial component, and explicit cues for the unattained location/reward limit the comparison to general features. The task used here also involves choice conflict between reward and effort. The ACC appears to play a role in resolving such conflicts (*Hillman and Bilkey, 2010*), and could account for the high frequency of excursions in the present data. If the excursions support the comparison of realized and fictive outcomes more generally, this suggests that the broad post-reward activation of neurons in the ACC and nearby structures observed in several species (*Amiez et al., 2006*; *Kennerley et al., 2006*; *Gruber et al., 2010*) may involve similar excursions and the recall of information in multiple modes.

Our present data extend previous work demonstrating that the activity of mPFC is sufficient to decode spatial position on a track (*Fujisawa et al., 2008*). Whether the encoded information is a

pure spatial signal or due to encoding of spatially-locked actions, stimuli, and/or events remains unclear. The task is repetitive and rats' movements tends to be stereotyped, so that specific actions (e.g., turns) and task events (e.g., approach to barrier) occur at the same location on every trial. Whatever its nature, our analysis reveals that the ensemble activity is sufficient to predict where an animal is with an unprecedented level of spatial resolution. It should be emphasized that this spatial information is multiplexed with many other task features encoded by this region, such as reward and effort, but can be extracted by an appropriately trained neural network decoder. This work extends on the work of others who have shown that the medial prefrontal region encodes a trajectory through task space (*Lapish et al., 2008*; *Durstewitz et al., 2010*) and shows that, at least under some circumstances, such a state-space trajectory is isomorphic with real world spatial coordinates.

Why is spatial encoding prevalent in the ACC? We propose that the ACC may form a topographically-organized representational space, based on real space or action/events at particular locations on the maze, which can serve as a scaffold for the encoding of behaviourally-relevant events. For example, if the rat is attacked by a neighbor near its nest, the ACC may encode the event and trigger avoidance on subsequent visits to that vicinity. This is similar to a recent proposal that the orbitofrontal cortex uses a map of abstract task-states to facilitate reinforcement-based behavioural adaptation (*Wilson et al., 2014*), except that in our case, the representation has real-world spatial correlates. Our proposal is also closely related to past proposals that the medial PFC likely forms and stores schema which map context and events onto appropriate actions (*Jung et al., 1998*; *Miller and Cohen, 2001*; *Alexander and Brown, 2011*), which serves to engage appropriate emotional or motoric responses to a given set of events in light of past experience (*Bechara and Damasio, 2005*; *Euston et al., 2012*). Again, the differentiating feature of our proposal is a higher spatial resolution. The spatial representation in mPFC often smoothly varies as an animal navigates the task space, but it can also drastically shift its response profile over the same task space in some circumstances, such as a switch of task rules (*Rich and Shapiro, 2009*; *Durstewitz et al., 2010*; *Ma et al., 2016*). These investigators proposed that this provides a shift in context so as to facilitate learning or utilizing different sets of associations (e.g. action-outcome). It remains unclear whether these shifts are due to a global remapping of the entire ensemble or only a subset of task-relevant cells. The decoding algorithms demonstrated here may be useful for determining if schemas (a.k.a. mental models or cognitive maps) retain associative information about space or other features across such shifts, or if ACC wipes the slate clean in some conditions.

The proposal of a mnemonic schema organized around position does not preclude its role in flexibly encoding other information to support decisions. Rather, it is a framework for the integration of information over several time scales, from consolidated memories to short-term 'working' memory, which is well supported by a large body of evidence in rodents and primates (*Euston et al., 2012*). The ACC thus uses information gleaned over both recent and remote experiences to form a model of the world organized around the spatial feature of the environment that also includes features useful for decisions that impact affective state, such as finding food and avoiding pain.

A novel aspect of the present data is the excursions from the present state at the choice feeders. This raises the possibility that the brain can mentally navigate the ACC map to recall information, or even generate hypothetical states consistent with the world model. Such prospection is consistent with the limited evidence available in other rodent PFC regions (*Steiner and Redish, 2014*), and is consistent with evidence in primates, which we discuss later. In our study, these shifts may have been due to encoding of (1) the spatial location of the alternate feeder, (2) the expected reward at that location and/or (3) the sensory features (e.g. proximity to a ramp) of the alternate location. These factors were partially confounded in our study. The mPFC is well known to encode reward amount (*Pratt and Mizumori, 2001*; *Kargo et al., 2007*; *Horst and Laubach, 2013*; *Insel and Barnes, 2015*) and may plausibly encode sensory features, but our evidence weighs in favor of a spatial shift. First, the excursion patterns were distinct from the patterns normally observed at the feeder, but overlapped with the patterns at the remote feeder. Second, excursions did not originate or terminate at the central feeder, even though the reward type and volume was comparable to that at the target feeders. Ultimately, whether the shifts are based on space, reward, or sensory features, our data still suggest that excursions involved a shift away from the present target feeder to encode features of the unselected target feeder, thus processing information related to choices and outcomes. The partial discrimination among excursion patterns may result from confounds of reward and locations, the similarity of sensory features of the two target feeders, or could reflect processing

of latent information (e.g. affective state). We also note that spontaneous reactivation of ensemble neural activity during replay events often differs in the number or timing of spikes as compared to the patterns during behaviour (*Foster and Wilson, 2006*; *Euston et al., 2007*). The excursions in our data were relatively brief as compared to the time of feeder occupancy, raising the possibility of temporal compression as observed during replay, and likely introducing additional confounds for the classification analysis. We made no attempt to optimize the pre-processing of the input signal, such as the smoothing kernel width or normalization/convolution, which likely would have partially accounted for these effects.

We speculate that the brain dynamics involved in the excursions are not isolated to the ACC, but are likely coordinated with that in other brain regions. The hippocampus sometimes also generates replay events after reward consumption (*Foster and Wilson, 2006*; *Carr et al., 2011*). These occur during large amplitude fluctuations of the field potential called sharp wave ripples, and occur in bouts lasting several hundred milliseconds. Task-related cells in the mPFC are modulated by these ripples following reward consumption, suggesting that this is a period of communication between the hippocampus and mPFC (*Jadhav et al., 2016*). A similar post-reward replay in the sensory domain has been reported in the orbitofrontal cortex (*Steiner and Redish, 2014*). Coordination of such events in ACC, orbitofrontal cortex, and hippocampus during pauses of directed behaviour following reinforcement would account nicely for the activity pattern and timing of the so-called default mode network (*Buckner et al., 2008*). Activation of this network in humans occurs during pauses in directed action, typically after reinforcement, and is associated with 'mind wandering' often involving imagined shifts in time and place (*Buckner and Carroll, 2007*). Analogous, and possibly homologous, default mode networks have been reported in non-human primates and rodents (*Hayden et al., 2009b*; *Mantini et al., 2011*; *Lu et al., 2012*). The ACC, hippocampus, and other structures comprising the telencephalon emerged early in vertebrate evolution hundreds of millions of years ago, and likely supported a predatory foraging habit (*Murray et al., 2017*). The widespread conservation of the telencephalon among modern vertebrates suggests that it has functions useful in many situations and natural environments. It is therefore possible that a proposed human homologue (area 24) of rodent ACC (*Uylings et al., 2003*; *Seamans et al., 2008*) may also employ spatial associations to organize and navigate a schematic world model (*Kaplan et al., 2017*). It may, therefore, not be a coincidence that space-based imagery is one of the most prevalent top-down mnemonic strategies employed by humans, which has been used throughout recorded human history (*O'Keefe and Nadel, 1978*). For instance, a person may imagine being in a particular restaurant in order to recall food options and quality, which is useful for making future dinner plans. The large expansion of granular prefrontal cortex, much of which connects extensively with ACC, likely endows primates with a greater ability to abstract problems (*Seamans et al., 2008*; *Murray et al., 2017*), and possibly a greater ability to exert top-down control over ACC dynamics. Therefore, if primate ACC has a schematic world model organized similarly to that shown here, the dynamics of excursions and any shifted perception associated with them are likely different than those in rodents. In other words, the neurophysiology that leads to excursions might be similar in rats and humans, but we make no claims that the prospective representation of information via excursions in rats is perceived or controlled similarly to prospection in humans. Along the same lines, the strong correlation of excursions with disfavour of a feeder is consistent with the activation of human ACC in regretful situations (*Coricelli et al., 2005*), but we have no independent means to assess if rats perceive regret in the present data.

We propose that the excursion events represent navigation of a schematic world model organized around spatial position for some purpose related to task performance, such as comparing the utility of the obtained reward to an unattained one. The emergence of excursions exclusively at the target feeders, and not the center feeder, suggests a role in outcome comparison or future choice. Excursions did not terminate at the center feeder, suggesting they do not encode the subsequent action from the target feeder, which is always a return to the center feeder as enforced by gates on the track. It is possible that the excursions reflect an unexecuted plan to move from the occupied feeder to the other. If this were the case, however, we would expect to occasionally observe excursions when the rat is at the center feeder or other location on the track. A possible alternative is that the excursions reflect a mechanism for shifting strategies. The rodent ACC is involved in shifting responses (*Joel et al., 1997*; *Birrell and Brown, 2000*), and appears to sustain information over time (*Dalley et al., 2004*; *Takehara-Nishiuchi and McNaughton, 2008*). Although speculative, it is

therefore possible that the excursions trigger a memory trace in ACC that promotes a response shift at the next visit to the choice point on the track. In other words, the ACC may have made a decision for the next choice while at the target feeder, which could preclude excursions at the center feeder or other intermediate point. The ACC is only one of several dissociated circuits that influence binary choice (*Gruber and McDonald, 2012*; *Gruber et al., 2015*), and is posited to bias competition among these other systems (*Murray et al., 2017*). Excursions may therefore have a probabilistic influence on future choice rather than fully determining it. The present data show only a correlation between revealed feeder preference and the likelihood of excursion. The design of the present task (e.g. forced alternation and relatively short blocks) prevents us from rigorously testing whether the excursion events influence future choice. We note that other evidence of spontaneous activation of task-related neural ensembles in the cortex and hippocampus has similarly shown correlation with general features of behaviour, such as learning, but most have not yet been shown to accurately predict future actions on a trial-by-trial basis (*Wilson and McNaughton, 1994*; *Euston et al., 2007*; *Fujisawa et al., 2008*; *Dragoi and Tonegawa, 2013*; *Steiner and Redish, 2014*; but see *Johnson and Redish, 2007*). We anticipate that advances in collecting and decoding ensemble neural activity will reveal such linkages between retrospective or prospective encoding and future behaviour.

## Materials and methods

### Behavioural apparatus

We constructed a running track 15 cm wide, with 36 cm high walls on both sides. It was configured into a 'figure-8' track measuring 102 cm long, 114 cm wide, and 60 cm height from the floor. Reward was delivered via three conical plastic feeders (24mm diameter). One was located on the central arm, and two others on 6 x 15 cm platforms at the north corners of the track. The reinforcement was a chocolate-flavored beverage (Ensure, Abbott laboratories, Brockville, ON). The platforms were elevated 0-48 cm above the track. The ascent to the platform was by a vertical wire mesh (1.6cm thick galvanized steel wire with a 1.25cm square spacing). The descent was by a ramp made from the same material, but with a solid opaque plastic immediately under the mesh to provide support. The elevation of each platform was independently controlled by a stepper motor (Model 23Y9, Anaheim Automation, Anaheim, CA) driven by a stepper motor controller (Model G251X, Gecko drive, Tustin CA). A rack and pinion gear system was used to carry the platform up and down. A programmable digital input/output board (National Instruments PCIe-7841R, Toronto, ON) and custom software written in Microsoft Visual Basic and Labview (National Instruments, Toronto, ON) were used to automatically control and store the time of track events.

### Data collection

We used Fisher-Brown Norway or Long-Evans rats born and raised on-site. Rats were habituated to handling for two weeks prior to the experiment. We surgically implanted a recording drive prior to any training. The drive and implantation were carried out as described previously (*Euston and McNaughton, 2006*). The position of the animal and neural signals were recorded simultaneously with a digital acquisition system (Cheetah SX, Neuralynx, Tucson, AZ). Neural signals were amplified with a unity gain headstage (HS-54, Neuralynx, Tucson, AZ), amplified with a gain of 1000, and band pass filtered between 600 and 6000 Hz. Voltage waveforms exceeding a manually set threshold were recorded during behaviour, and then sorted into distinct clusters offline.

Animals began training on the figure-8 track following a one-week recovery from surgery. Behaviour was shaped by allowing rats to navigate the track for 7-10 days with no variation of reward volume or barrier height. All subsequent sessions followed a fixed reward/effort schedule in which the task was organized into 6 blocks of 16 trials. Gates on the track forced the rat to alternate between the left and right loops on the first 10 trials of the block. The rat was free to choose either side for the remaining 6 trials of the block. The barrier height (0-46 cm) and/or the reward volume (30 or 120 uL of chocolate beverage) at one or both of the feeders changed across each block. The block order was: [S0, B0], [S0, B1], [S0, B2], [S2, B2], [B0, S0], [B2, S0], where the letter indicates reward volume (S for small, B for big), and the number codes the relative effort. The block sequence is repeated until the animal stops performing trials. The side with initially large reward is counterbalanced over

consecutive sessions. Animals were reinforced at the central feeder on every lap with the same chocolate beverage and small reward volume as for the choice feeders.

## Decoding ensemble activity

We first removed individual trials with durations longer than 1.5 times the median trial length in each session so as to reduce neural correlates of non-task behavior (e.g. grooming, rearing) in the data. We further eliminated neurons with an average firing rate below 0.5Hz because our initial tests indicated that these cells did not improve decoding accuracy. We then used a Gaussian kernel with a standard deviation of 150 ms to smooth the spike data, and then binned the resultant signal by 50 ms. The position of the animal assigned to each bin was the average of the coordinates of the video tracker system for the corresponding time window. We next applied the square root transformation to the binned firing rates to normalize the activity distribution (e.g. make them more Gaussian) among neurons. We then used the z-transform so that the activity of each neuron would have zero mean and unit variance. For decoding using the Bayesian method, we discretized space into regions of 4 cm by 3 cm. The dANN method operated at the pixel resolution of the video tracker (0.27 x 0.24 cm). To ensure that the difference in spatial resolution among decoding methods did not present confounds, we applied the dANN method to the data discretized in the same way as for the Bayesian method. This did not affect the prediction error, so we only show the results from the finer resolution in the figures.

The position decoder was a multi-layer feedforward artificial neural network with three hidden layers. The number of units in the input layer equals the number of recorded neurons in each session. The number of units in the first, second, and third hidden layer is 100, 50, and 25, respectively. The output layer consists of two units, which represent the coordinates of the position on the track. The activation function of the first two hidden layers is rectified linear, and the activation function of the third layer is hyperbolic tangent. The number of units and the activation function for each layer were selected via cross validation. The quality of predictions was quantified with the mean squared error (MSE) between the actual position of the rat and the predicted location:

$$MSE = \frac{1}{2N} \sum_{i=1}^{N} \left\| \begin{bmatrix} \hat{x}_i \\ \hat{y}_i \end{bmatrix} - \begin{bmatrix} x_i \\ y_i \end{bmatrix} \right\|^2$$

where $\hat{x}_i$ and $\hat{y}_i$ are the predicted coordinates for the $i$-th test sample, and $x_i$ and $y_i$ are the target coordinates. $\| \ \|$ represents $L^2$-norm, $N$ and is the number of samples in the test set. For the Bayesian decoder, we computed the MSE between the center of the target region and the predicted region.

We used mini-batch gradient descent with weight decay and momentum to train the network. The batch size was 100 and the learning was stopped after 100 epochs. For each session, we created 10 sets of trials, each consisting of an equal number of left and right choice trials. We used 75% of the trials in each set to train the network and the remaining 25% to evaluate the model. The reported prediction error for each session was the average of the errors computed for each of the 10 test sets.

## Downsampling and greedy selection

To compute the reconstruction error versus the number of cells in the data, we created different datasets, each containing a subset of cells. The number of cells in these sets ranged from 4 to 40 by increments of 4. For each set size, we created 5 different datasets by sampling cells randomly. These steps resulted in 50 datasets, consisting of 10 different set sizes and 5 different cell sets for each size. The process for evaluating each set is the same as that described above. In order to determine the best set of cells for the decoding, we used the forward feature selection algorithm to choose the best set of 20 cells for decoding position. In this approach, the best set of cells is initially empty. We iterated over all the cells to find the cell that will result in the lowest error if is added to the current best set of cells. This cell is then added to the current best set. These steps are repeated until the best set reaches the desired cardinality. In order to determine the error for each candidate set, we used the same procedure described in the previous section (average of 10 sets of randomly selected trials).

## Error map construction

For each session, we created 10 different datasets, consisting of an equal number of right and left choice trials. We used 75% of the trials in each dataset as the training set and the remaining 25% as the test set. After training the network on each training set, we decoded the location of the rat for the corresponding test set. In the next step, we discretized the space into tiles of 4 by 3 cm, and for each test trial computed the maximum reconstruction error when the rat was in a particular tile. The error map for each session was constructed by taking the average of the maximum reconstruction error values over trials. The final map is constructed by averaging error maps of all sessions.

## Determining the reward and choice encoding cells

To determine which cells encoded information about reward or choice, we computed the average firing rate of all cells per trial in a 1.5 s window beginning immediately after reward delivery at the north feeders, so as to create an $m$ by $n$ matrix, where $m$ is the number of cells and $n$ is the number of trials. The target associated with each column of this matrix was a binary value that represented different conditions of the parameter of interest (right/left for choice, high/low for reward). For each parameter of interest, we created 50 random sets of trials, each one containing an equal number of trials with different conditions. We used 75% of the trials in each set as the training data and the remaining 25% as the test data. We fit a logistic regression model with norm-1 regularization (Lasso) and a maximum degree of freedom of 20 to each training set. Models with different degrees of freedom were trained using 5-fold cross-validation. For each set, we picked the model that resulted in the best prediction accuracy of the target based on the neural vectors, and then logged the cells that were assigned non-zero weights (e.g. they provide useful information for discrimination). To ensure that the selected cells were meaningful, we evaluated the accuracy of the model on the test data. In our experiments, the accuracy of the final model was always above 90%. Finally, we computed the percentage of times (out of the 50 sets) each cell was assigned a non-zero weight. For a value less than or equal to 30%, the cell was classified as not informative. For values greater than 30% and less than 70%, the cell was classified as relatively informative. For values greater than 70%, the cell was classified as very informative.

## Excursion detection

We increased the temporal resolution by reducing the smoothing kernel width to 120 ms and binning the resultant signal by 20 ms. All other features of the decoding network were as described above. For each session, we created 10 sets of trials by randomly selecting an equal number of left and right trials. The set size for each direction was half the minimum number of trials performed on either side. For each set, we divided the samples into two groups. The first one contained data points that fell into a time window of 1.5s after reward delivery in the selected trials. All the other samples from the set were assigned to the second group. We used the samples in the second group to train the decoder network, and then used the network to predict the location of the rat from the samples in the first group. We marked each test trial as a excursion trial if the maximum error between the decoded position and the actual location of the rat was greater than 70 cm during that trial. The statistics we report in this paper were computed by taking the average of the results we obtained on each set of test trials.

## Identifying excursion-related cells

We divided the samples in each test set created for excursion detection into two categories. The first category contained all excursion samples that fell into a window of 80 ms, centered at time at which the distance between the actual location of the rat and the predicted location was at its peak. The second category contained all the other samples. We computed the average firing rate of the cells per category, and applied a square root transformation to convert the activity distribution into a normal distribution. We then used a t-test with a significance level of 0.01 to detect the cells that had significantly different firing rates under the two conditions. We computed the percentage of time each cell was selected among cells with significantly different firing rate and used the same procedure described above to classify cells into three levels of importance.

## Adding noise, shuffling data, and constructing confusion matrices

Confusion matrices show the probabilities of assigning patterns to correct and incorrect categories. We preprocessed neural data as for the detection of excursions. The spike data was smoothed for each session using a Gaussian kernel of width 120 ms, and then binned by 20 ms. We selected two sets of data from each trial of the task: the set containing all the samples that fell into a window of 750 ms after the off-set of the center feeder; and the set of all data points that fell into a window of 1500 ms after the off-set of the selected side feeder. Each sample was labeled based on the feeder associated with it (right/left/center).

To create a noisy dataset, we concatenated the sets associated with each feeder and computed the standard deviation of the inter-spike intervals of each neuron for each feeder. Then, we shifted spike times in each feeder set by values that were randomly drawn from Gaussian distribution with zero mean and a variance equal to 25% of the cell's firing interval variance in that set.

To create the shuffled dataset, we concatenated all the selected time windows for all the feeders and computed the inter-spike intervals for each neuron. Then we created a new spike time-series by randomly permuting the inter-spike intervals. The label of each time-point in the shuffled dataset was the same as the label of that time point in the original dataset.

To obtain a confusion matrix for each dataset, we performed the following procedure: we randomly selected an equal number of left and right trials and used the samples from the selected trials in the original data set to train a neural network classifier. The architecture of the classifier was identical to the network that was used for decoding position, except for the output layer, which had 3 units (to represent the three feeders) and a softmax activation function. For each dataset (original/ noisy/ shuffled) we used the samples from the remaining trials as the test set and computed the normalized confusion matrix. We repeated this process 10 times for each session, and computed the average classification accuracy of these repetitions. The reported results represent the average over all sessions.

## Discrimination analysis of excursion patterns

From each dataset created for excursion detection, we created a new dataset by performing the following steps. For the trials without excursion, we computed the mean firing rate of each neuron during a window of 0-1.5 s after the feeder was closed. For the trials with excursion, we computed the average firing rate of the neurons in a window of 100 ms, centered at the time at which the excursion was at maximum departure from the actual location of the rat. Then, we assigned a label to each one of these vectors: normal pattern at feeder A (A), excursion pattern at feeder A (A'), normal pattern at feeder B (B), excursion pattern at feeder B (B'). To differentiate between normal and excursion patterns at each feeder (A vs A' and B vs B'), we used a decision tree with a maximum number of splits of 10. To separate the excursion patterns at different feeders, we used an SVM classifier with a linear kernel. Because the number of samples was unequal among the classes, we used the area under the curve (AUC) of the receiver operator characteristic (ROC) as a measure of pattern separability.

## Additional information

### Funding

| Funder | Author |
| --- | --- |
| Alberta Innovates - Health Solutions | Bruce L McNaughton<br>David Euston<br>Aaron J Gruber |
| Natural Sciences and Engineering Research Council of Canada | Saeedeh Hashemniayetorshizi<br>Bruce L McNaughton<br>David Euston<br>Aaron J Gruber |

The funders had no role in study design, data collection and interpretation, or the decision to submit the work for publication.

## Author contributions

Ali Mashhoori, Conceptualization, Formal analysis, Visualization, Methodology, Developed the neural network algorithm under supervision of AJG, Analyzed the data, Discovered the excursion phenomenon, Produced figure panels, Extensively edited and/or commented on manuscript; Saeedeh Hashemnia, Data curation, Investigation, Data collection and pre-processing, Extensively edited and/or commented on manuscript; Bruce L McNaughton, Writing—review and editing, Extensively edited and/or commented on manuscript; David R Euston, Resources, Supervision, Methodology, Project administration, Designed the task, built the apparatus, and performed the surgeries, Supervised data collection and pre-processing by a technician, Extensively edited and/or commented on manuscript; Aaron J Gruber, Conceptualization, Formal analysis, Supervision, Visualization, Methodology, Writing—original draft, Project administration, Supervised the development of the neural network algorithm by AM, Analyzed the data, Discovered the excursion phenomenon, Produced the figures, Wrote the initial manuscript and extensively edited and/or commented on it

## Author ORCIDs

Saeedeh Hashemnia (iD) http://orcid.org/0000-0001-8158-3145
Aaron J Gruber (iD) http://orcid.org/0000-0003-2700-5429

## Ethics

Animal experimentation: All procedures were approved by the university's animal welfare committee (Protocol #1512) in accordance with the Canadian Council on Animal Care.

## Decision letter and Author response

Decision letter https://doi.org/10.7554/eLife.29793.sa1
Author response https://doi.org/10.7554/eLife.29793.sa2

## Additional files

### Supplementary files

• Transparent reporting form

### Data availability

The pre-processed data used for the paper and the computer codes for the artificial neural network are available for download at a publicly accessible repository (https://github.com/mashhoori/ACC-Recalls-Features-of-Remote-Reward-Locations; copy archived at https://github.com/elifesciences-publications/ACC-Recalls-Features-of-Remote-Reward-Locations).

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
