## [Decision Letter]

Thank you for submitting your article "Anterior cingulate cortex mentally teleports to preferred reward locations after reinforcements elsewhere" for consideration by *eLife*. Your article has been reviewed by two peer reviewers, and the evaluation has been overseen by Timothy Behrens as Reviewing and Senior Editor. The reviewers have opted to remain anonymous.

After discussion between the reviewers and myself, we would like to invite a revision of the manuscript. As you will see below, though, there are some criticisms of the decoding which are pretty central to the message of the paper, so it is very important that they are dealt with carefully and clearly before we could consider publication.

Summary:

The manuscript contains two novel sets of findings in an effort/reward maze task. First that spatial location can be decoded from ACC/mPFC ensembles. Second that when an error is made, and the animal discovers they are at the wrong feeder, the ensemble encodes the location of the alternative (rewarded) needed. This second finding is particularly interesting as it sits within a literature that implicates ACC activity in behavioural change but delivers new detail about the content of the activity that promotes this change in behaviour. The reviewers (and editor) were therefore enthusiastic about the findings.

Essential revisions:

The first set of essential revisions pertain to the critical decoding of the alternative feeder.

1a) The main issue relates to problems with the way the decoding results are interpreted. The authors have not done sufficient work to demonstrate that the 'teleportation' effect is not an artifact of decoding errors given the way the cells respond around the feeders. The predictions are worse around the feeders, even when the rats are at feeders. Firing around the feeders is more spatially diffuse and from the examples, cells respond at multiple feeder locations. Teleportation is a decoding error. Given these properties of the neurons, the most common type of decoding error around a feeder will be the other feeder location. As far as I can tell these are not switches between feeder-specific patterns, but it is simply that if you force a decision, the most likely error will be around the other feeder.

1b) Another type of error that I am less concerned about is that because the neurons seem to fire more around feeders, the neural net could interpret any elevation in firing as a feeder location, if decisions are forced. This is probably not so much of an issue, but it again illustrates the main problem that the reader really has no idea how the errors occur.

1 overall) To make a teleportation claim, the authors would need to show that: (1) The two feeder locations are associated with statistically different ensemble patterns. These need to be evaluated relative to all the other patterns associated with all the other location specific patterns as well as various shuffled patterns. (2) That there is switching between the two statistically different feeder patterns after reward. (3) That there is a relative absence of switching to the other statistically different patterns that are associated with all the other non-rewarded maze locations.

There were also some more minor concerns about the decoding of spatial location.

2) On the other hand, the fine-grained encoding of spatial location is quite remarkable for mPFC ensembles. This finding seems less problematic, although it would also benefit from a more rigorous approach. How good is decoding on a lap by lap basis? Is decoding so good because the feeder representations are so diffuse relative to the size of the maze? The authors should comment on whether they would expect such good decoding in the absence of feeders or on a much larger maze.

The second set of concerns relates to how the ACC decoding relates to behave.

3) Second, I found it hard to work out exactly how the rats were performing and how the analyses related precisely to what the animals were doing. There is a description of the different blocks (Figure 1B) and the mention of "choice biases" (Introduction), but for instance no basic description of overall performance. How many trials / sessions were there for these rats? What choices did they make and how consistent was their choice performance across the session? And how does this relate to the decoding accuracy? For instance, on choice trials, did the alternate feeder decoding accuracy correlate with performance on trial t+1? Are there differences across the block of forced trials when there is a change of value that also results in a change in preference? Other studies (e.g., Hillman and Bilkey) have suggested that ACC cells only care about situations where there is cost-benefit conflict – i.e., there is a need to overcome greater cost to achieve a greater reward.

4) I think the data in Figure 3C might be really interesting, but I couldn't work out at all what the 7 depicted points actually were. Similarly, it is described in the text that "many cells involved in the excursion also encoded reward value and/or choice" – but it is not clear what counts as "many" and the division into "relatively" or "very" important seemed arbitrary and unnecessary. There are also lots of passing mentions of task elements or analyses that are not described. For instance, there is mention of a central feeder, but I couldn't find a description of how much reward was delivered there or even whether it was delivered on every trial. This is important as those control analyses (e.g., Figure 3B, Introduction) rely on this feeder being an equivalently salient location where presumably the animal also pauses to consume the reward.

5) In general, tightening up the link between the ACC decoding and what the rats were doing would potentially really strengthen the findings.

Lastly, there was a general feeling from the reviewers and editor that the manuscript should be written in a more sober tone. This is most clearly expressed in the following 2 comments.

6) I felt the framing of the whole paper was unnecessarily hyperbolic. The first sentence of the Abstract states that "Humans have known for thousands of years that the recall of multimodal information can be greatly facilitated by imagining the location associated with it". Thousands of years? What does this actually mean? Then the first main paragraph (Introduction) describes literature on ACC, the control of memory retrieval, cost-benefit outcomes, regret and particularly the function of the default mode network. However, this doesn't really line up. The medial frontal focus of the default mode is generally in rostral and ventromedial parts of medial frontal cortex (the recording site here looks about the least activated part of the rat default mode in the Lu et al., 2012 paper), the reward-guided literature is focused mainly on dorsal ACC, and the cited memory retrieval paper examines an area that looks to me like adjacent secondary motor cortex rather than ACC proper. Also, the emphasis on "regret" and "regretful situations" really isn't supported by any rigorous evidence. Just because the effect occurs on free choice trials when they choose the on average less preferred option does not automatically confer that the rats must be regretting the choice, particularly as the effect is still present on forced choice trials where there was no other option.

What is particularly frustrating about this is that I don't see any reason to try to oversell the finding by surrounding it in this vague and overblown terminology. The basic effect – of representing a potential alternative outcome at the time of reward collection – is very neat and fits well with the wider literature of this region in encoding the value of doing alternative courses of action and updating internal models of the environment.

[Editors' note: further revisions were requested prior to acceptance, as described below.]

Thank you for submitting your article "Anterior cingulate cortex mentally teleports to preferred reward locations after reinforcements elsewhere" for consideration by *eLife*. Your article has been reviewed by two peer reviewers, and the evaluation has been overseen by a Reviewing Editor and Timothy Behrens as the Senior Editor. The reviewers have opted to remain anonymous.

The reviewers have discussed the reviews with one another and the Reviewing Editor has drafted this decision to help you prepare a revised submission.

I'm afraid the key point that the reviewers raised last time is not well addressed in the eyes of the reviewers. It is still unclear whether the decoding to the new feeder position is driven by the activity that is common between the two feeders. Reviewer 2 goes into some detail about this point below and I have read through the argument and it seems a strong one to me. Below, reviewer 2 has suggested a number of tests that would be more convincing. In discussion we thought the fairest thing was to give you a final chance to address this point.

Reviewer #1:

I think the revised manuscript is more tightly focused and improved. The findings are potentially very interesting.

While I understand that the authors would have liked to make a stronger connection between behaviour and physiology but maybe cannot given the task design, I would nonetheless like just a bit more data just to show that the animals are performing the task at all as you'd expect. For instance, it's stated that the rats chose one option on ~88% of free choice trials, but it's not as far as I could see stated what these choices were. This is important given the high reward side switches at block 4 in Figure 1B and also there are some blocks where it is straightforward to determine what to do (HR/LE v LR/LE or LR/HE v HR/HE) and others where the contingencies are mixed. Could a summary of the behavioural performance in the 2 rats / 7 sessions analysed for teleportation events be included as part of Figure 1 perhaps?

I also still was a bit confused by what is now Figure 4A. If the data are the averages from each of the 7 sessions, what is "the feeder" (as in "preference for the feeder in free-choice trials" on the x-axis in 4A)? And why, if the data are averages, are there some sessions with very strong/weak preferences for "the feeder" (i.e., < 25% or > 75%) when the animals always had a strong preference in a block? Does that mean in some sessions animals were seldom choosing the other feeder at all in spite of the difference in values?

Reviewer #3:

The authors did not adequately address my concerns from the last round. The main point is that the two feeder locations are different in that they have different x-y values but similar in that they both deliver reward and the neurons encode both sources of information. That means that if you set up an analysis to search for differences (e.g. LDA), the activity around the feeders will appear to be different. On the other hand, the most common decoding errors ('teleportations') will reflect the similarities. The new controls they provided do not get at this issue.

1) In response to my critiques, the authors show a single LDA panel. LDA finds the most discriminating axes and since the authors showed that the networks strongly encode spatial information, LDA presumably separated the 3 feeder locations based on location-related activity. The LDA figure is therefore just another demonstration of what they already showed in Figure 2. The new controls are of little value (see point 3). The authors need to show that the isolated location representations actually shift independent from the contaminating influence of the common reward-related representations. Admittedly, this will be difficult because (in the Introduction) "Our analysis revealed that many cells involved in the excursion also encoded reward value and/or choice information". If single neurons or the network as a whole encodes both spatial and reward information, the dNN will learn this. As a result, a decoding error is not really an error or a teleportation but the dNN indicating that part of the input is reward related and that part of the input is common to the two feeders. The relative degrees to which spatial and reward information is present likely varies from trial to trial and "teleportations" may be the trials where there is a more reward-like input pattern than a location-like input pattern. The authors therefore need to come up with clever ways to parse the spatial and reward signals. Since single neurons encode both reward and location, extracting neurons is not a solution. Instead, the authors could potentially use the LDA for this. Assuming the LDA axes in Figure 3 are oriented to maximally separate activity based on location (please check if this is correct), activity on 'teleportation' trials could be projected onto these axes. The points from the blue feeder trials should then fall into the red cluster and vice versa. Furthermore, the likelihood of finding a 'blue' point in the red cluster on teleportation trials should be much higher than the likelihood of finding a non-feeder point in the red cluster. If the authors do not like this solution, they could implement some other means to first parse out the contribution of reward encoding before using the dNN or to show that just the isolated spatial representations shifts while the rat is stationary at a feeder.

Also, as I asked in the last round, please show a complete error map across the entire maze at a spatial resolution like that shown in Figure 2. If the rat is at the feeder on a test trial the distribution of predicted locations should be localized around the other feeder. Again, this is probably because the dNN is picking up on common reward-related activity, but this map would still be useful.

2) I still have some trouble with the precision of the spatial encoding, given the limited training data. Since these neurons have low firing rates and are inconsistent, with such limited training data most of the ~5cm locations that were accurately decoded would never be associated with any activity, let alone consistent activity. The precise spatial encoding may instead be an artifact of how the spiking data was pre-processed. In subsection “Decoding ensemble activity” the authors state that they smoothed the data with a kernel having a s.d. of 150ms and then binned at 50ms prior to performing a square root transform. This would smear out the temporal and spatial inconsistencies in the raw data. By eliminating the inherent trial-to-trial and bin-to-bin inconsistencies and filling-in sections of the maze where no activity was present, the pre-processing may have essentially created a highly continuous and repeatable activity mosaic across the maze. Please repeat the spatial decoding analyses using the raw binned data.

3) I do not understand the 'similarity' controls. The noise procedures are of limited value since any information on this timescale is completely occluded by the pre-processing. Second, it begs the question of how much noise is useful to make the point? Too much and all decoding would break down, too little and the dNN would be robust to it. Again, the critical control is to extract just the spatial representation components and show that they shift back and forth when the rat is at one feeder.

4) In the Introduction the authors discuss the impact of preferences because of unequal effort-reward utilities. First, I am unclear how the unique blocks (subsection “Data collection”) and forced/free trials impacted the selection of training/test trials. If the reward magnitudes do not match there would be less excursions given the decreased (although still not zero) similarity in the reward representations. Second, Figure 4A would only be valid if the neurons and the dNN had the same exposure to the two feeders for the same reason that more training data improves classification.

[Editors' note: further revisions were requested prior to acceptance, as described below.]

Thank you for resubmitting your work entitled "Rat anterior cingulate cortex recalls features of remote reward locations after disfavoured reinforcements" for further consideration at *eLife*. Your revised article has been favorably evaluated by Timothy Behrens (Senior editor), a Reviewing editor, and one reviewer.

Tim Behrens sent this back to reviewer 3 and looked at it himself. Both agree the manuscript is much improved, but there are a few remaining changes that we would like you to make, detailed below.

1) Soften the language about a lack of similarity between feeder representations. The Abstract highlights the multi-plexing of reward and position representations and Figure 4 shows there are a separable reward and position components during excursions. Yet the Results are written as if to completely disregard this possibility. For example:

Subsection “Excursions of spatial encoding from the physical position to a feeder”: These analyses do not provide evidence that the excursions are not due to similarity at the feeders. LDA only finds the most discriminating axes and says nothing about other dimensions in which the feeder representations could be similar.

Subsection “Excursions of spatial encoding from the physical position to a feeder”: An AUC of 0.75 does not provide convincing evidence "that the excursion patterns are not generated from a common state" in spite of the classifier "that they sometimes have overlapping features". Also, Figure 3H is not helpful and needs to be replaced with actual data.

In these places and throughout the manuscript (e.g. Discussion section) the language about a lack of similar components at the two feeders needs to be softened. It is ok that there are similarities because these similarities cannot completely explain the excursions.

2) The discussion of shifts and remapping in the Discussion section is confusing. Isn't the situation the same as in the present study whereby there are changes in the ensemble that vary in magnitude across cells? I am not sure how dimensionality reduction plays into all this. Furthermore, Rich and Shapiro used a correlation matrix of ensemble activity on different trials, which is not really a dimensionality reduction technique. Likewise, Ma states: "Multivariate analyses were always performed in the full multidimensional space, but for the purpose of visualization, N-dimensional population vectors were projected down into a 3D space by means of metric multidimensional scaling." I have no idea what the authors are trying to say in this section.

3) There should be some discussion about why there are no excursions to the central feeder on non-preferred trials.

---

## [Author Response]

Summary:The manuscript contains two novel sets of findings in an effort/reward maze task. First that spatial location can be decoded from ACC/mPFC ensembles. Second that when an error is made, and the animal discovers they are at the wrong feeder, the ensemble encodes the location of the alternative (rewarded) needed. This second finding is particularly interesting as it sits within a literature that implicates ACC activity in behavioural change but delivers new detail about the content of the activity that promotes this change in behaviour. The reviewers (and editor) were therefore enthusiastic about the findings.Essential revisions:The first set of essential revisions pertain to the critical decoding of the alternative feeder.1a) The main issue relates to problems with the way the decoding results are interpreted. The authors have not done sufficient work to demonstrate that the 'teleportation' effect is not an artifact of decoding errors given the way the cells respond around the feeders. The predictions are worse around the feeders, even when the rats are at feeders. Firing around the feeders is more spatially diffuse and from the examples, cells respond at multiple feeder locations. Teleportation is a decoding error. Given these properties of the neurons, the most common type of decoding error around a feeder will be the other feeder location. As far as I can tell these are not switches between feeder-specific patterns, but it is simply that if you force a decision, the most likely error will be around the other feeder.1b) Another type of error that I am less concerned about is that because the neurons seem to fire more around feeders, the neural net could interpret any elevation in firing as a feeder location, if decisions are forced. This is probably not so much of an issue, but it again illustrates the main problem that the reader really has no idea how the errors occur.1 overall) To make a teleportation claim, the authors would need to show that: (1) The two feeder locations are associated with statistically different ensemble patterns. These need to be evaluated relative to all the other patterns associated with all the other location specific patterns as well as various shuffled patterns. (2) That there is switching between the two statistically different feeder patterns after reward. (3) That there is a relative absence of switching to the other statistically different patterns that are associated with all the other non-rewarded maze locations.

This is an excellent point, and we apologize for not addressing it adequately in the first draft. This has been our primary concern for the past 18 months since we first found the teleportations. We have undertaken several approaches to convince ourselves that the phenomenon is real and not a methodological artifact. We now include this evidence so as to support our claim that this is a neurobiological phenomenon. I would like to first point out that I suspect that our initial presentation of the mean MAXIMUM error was causing confusion and/or skepticism. We have heavily edited the manuscript and added figure panels to present the evidence as follows:

1) The mean and median prediction error is very good across the entire maze – feeder and non-feeder alike. We show this now in two panels. Figure 2F shows all of the ‘instantaneous’ (50ms) errors as a function of linearized position to illustrate that the mean accuracy is very good at the feeders. It is the distribution of the outliers that is unique at the feeders. This can be seen in the one session data (2F), and the new population averaged data in 2G.

2) We now show (new figure panel 3E) that linear discriminate analysis (LDA) produces clusters of the neural patterns recorded from the feeders. This is graphical evidence that is independent from our decoding network. This shows that linear combinations of neural activity can separate the patterns.

3) LDA is linear and could miss non-linear interactions that contain information. In order to not constrain methods to linear combinations, we used a 3-layer neural network to classify the patterns of activity at the feeders. We then quantified the classification error for all possible combinations. This resulting plot of classifications is commonly called a ‘confusion matrix’, which we show in Figure 3F. We show a statistical comparison of these summed errors in panel Figure 3G. This shows that the accuracy is very high in the original data, and that moderate noise by spike shuffling does not decrease the accuracy whereas full shuffling does drastically decrease accuracy. If the teleportation events were due to similarity of patterned neural activity, this shuffling should cause an increase in the number of errors.

4) To recap, there are 4 new key pieces of evidence in the manuscript: (1) the dANN clearly separates the patterns at feeders by assigning them to well-separated spatial positions, (2) the LDA shows pattern separation, (3) a separate classifier network also shows excellent accuracy, and (4) moderate levels of spike shuffling do not increase errors. A fifth piece of evidence is the fact that the teleportations occur over several time bins and last longer than the smoothing kernel, which would be exceedingly unlikely to occur by chance. This evidence is outlined in a new paragraph in the Introduction.

There were also some more minor concerns about the decoding of spatial location.2) On the other hand, the fine-grained encoding of spatial location is quite remarkable for mPFC ensembles. This finding seems less problematic, although it would also benefit from a more rigorous approach. How good is decoding on a lap by lap basis? Is decoding so good because the feeder representations are so diffuse relative to the size of the maze? The authors should comment on whether they would expect such good decoding in the absence of feeders or on a much larger maze.

Thank you for pointing this out. We now show all errors for one session (Figure 2F). We had already shown the prediction error for two laps of the task (Figure 2C). We provide some evidence that these are representative laps by indicating the prediction error of these two laps on the cumulative distribution of error – the blue lap is slightly less than the median, and the black is somewhat higher than the median (because of the log scale). We also now show the mean prediction error across all sessions in Figure 2G. We hope that this reduces confusion about the maximal error, shown in the lower graphic in this panel. The maximum is averaging the outliers, as seen in Figure 2F. These data show that the goodness of prediction is independent of location and occupancy of position (because the plots are normalized by time and space). So, the prediction accuracy is not strongly influenced by the rats’ occupancy of the feeder zones.

We further now report that adding noise to the spike times has a small effect on the accuracy of the position decoding (Introduction) and show the data in Figure 2C. This shows that the encoder is robust against noise. We want to point out that we have taken pains to cross-validate all of the decoding models by training and testing on separate portions of the data so as to avoid ‘overfitting’. This part of our approach is what allows the network to be robust against noise.

The second set of concerns relates to how the ACC decoding relates to behave.3) Second, I found it hard to work out exactly how the rats were performing and how the analyses related precisely to what the animals were doing. There is a description of the different blocks (Figure 1B) and the mention of "choice biases" (Introduction), but for instance no basic description of overall performance. How many trials / sessions were there for these rats? What choices did they make and how consistent was their choice performance across the session? And how does this relate to the decoding accuracy? For instance, on choice trials, did the alternate feeder decoding accuracy correlate with performance on trial t+1? Are there differences across the block of forced trials when there is a change of value that also results in a change in preference? Other studies (e.g., Hillman and Bilkey) have suggested that ACC cells only care about situations where there is cost-benefit conflict – i.e., there is a need to overcome greater cost to achieve a greater reward.

We have now included quantitative detail on the behavioural performance – particularly the number of trials (Introduction) and the bias of the rats (Introduction).

The second part of this comment hits on a potentially important aspect of the phenomenon- how does it influence behaviour. We tried to answer this question, but the design of the task presents insurmountable problems. In short, the task was not designed to test this. Roughly two thirds of the trials are forced alternating choice and can be fully predicted by the rat. This raises the possibility of spurious correlations in trial-by-trial analysis. For instance, on two thirds of the trials, the current neural activity is highly correlated with the current choice, the past choice, and the future choices (because of the alternation in forced choice). We did analyze whether neural activity was predictive of past or future choice, reward or effort, and found that it was as shown in Author response image 1:

However, we are not confident that these correlations would hold under a different choice schedule. Furthermore, the rats were descending a ramp on the side arms of the track when the platform was elevated, so some of the effort encoding could be related to mortoric differences in navigating different declines. For these reasons, we omitted neural correlates across forced-choice trials.

The 10 forced-choice trials were followed by 6 free-choice trials, in which the rat had very strong feeder preference. This led to very few trials in which the rat freely chose the less-preferred option. We therefore do not have sufficient data to address whether the teleportation is indicative of future behaviour. This is a very important question, and the focus of our ongoing experiments.

4) I think the data in Figure 3C might be really interesting, but I couldn't work out at all what the 7 depicted points actually were. Similarly, it is described in the text that "many cells involved in the excursion also encoded reward value and/or choice" – but it is not clear what counts as "many" and the division into "relatively" or "very" important seemed arbitrary and unnecessary. There are also lots of passing mentions of task elements or analyses that are not described. For instance, there is mention of a central feeder, but I couldn't find a description of how much reward was delivered there or even whether it was delivered on every trial. This is important as those control analyses (e.g., Figure 3B, Introduction) rely on this feeder being an equivalently salient location where presumably the animal also pauses to consume the reward.

The figure panel in question (now Figure 4A in the revised manuscript) is important. Each data point is the average of data from one session. It shows that teleportation events are more likely to occur at one feeder, and that this bias negatively correlates with the revealed preference for that feeder (the percentage of times the rat chose that feeder in free-choice trials). We realize that the symbolic representation in axis labels were difficult to understand, and so have relabeled them with descriptions that are hopefully more understandable.

We also now describe the central feeder in the main text, and state that rats get the small reward every time at this feeder. The reward saliency of this feeder should be on par with the choice feeders.

The determination of whether cells are important for conveying information about reward value and/or choice is difficult to quantify with methods commonly used in neuroscience. The teleportation events are very brief, we only have two levels of choice and reward, and each neuron can encode several variables (or their interaction). Moreover, the low number of trials relative to the variance of activation is not well-suited to linear models such as ANOVA. We therefore undertook a bootstrapping approach (which is a common method for determining confidence intervals) to determine how often a particular cell would be selected as an informative input by a classification algorithm. A better approach would be to test all combinations of cells in order to identify which were needed for the discrimination and the relative information contained by each. This is not computationally tractable, and so we report the frequency that each cell is selected. This is a common approach in machine learning. Nonetheless, the categories are indeed arbitrary. The obvious approach is to plot the actual proportion of trials that each cell was selected, but it then becomes very difficult to see which are jointly encoding information. We therefore left the color scheme discretized into three levels but change the description in the caption and the text to indicate that we are showing how ‘informative’ the cell is for discriminating the feature of interest, rather than its ‘importance’. The threshold for the three levels is arbitrary, but the results are not qualitatively affected very strongly by assigning different thresholds. We do not make any claims as to how frequently neurons jointly encode features, and therefore argue that our approach reasonable.

5) In general, tightening up the link between the ACC decoding and what the rats were doing would potentially really strengthen the findings.

We very much agree. We have done this in the spatial domain, but as stated above, our hands are tied by the choice schedule (forced alternation). We can indeed find strong correlations, but we are not able to parse out what is because of the deterministic nature of two thirds of the trials (forced-choice), and the strong bias in choices in the remaining trials (free-choice).

Lastly, there was a general feeling from the reviewers and editor that the manuscript should be written in a more sober tone. This is most clearly expressed in the following 2 comments.6) I felt the framing of the whole paper was unnecessarily hyperbolic. The first sentence of the Abstract states that "Humans have known for thousands of years that the recall of multimodal information can be greatly facilitated by imagining the location associated with it". Thousands of years? What does this actually mean? Then the first main paragraph (Introduction) describes literature on ACC, the control of memory retrieval, cost-benefit outcomes, regret and particularly the function of the default mode network. However, this doesn't really line up. The medial frontal focus of the default mode is generally in rostral and ventromedial parts of medial frontal cortex (the recording site here looks about the least activated part of the rat default mode in the Lu et al., 2012 paper), the reward-guided literature is focused mainly on dorsal ACC, and the cited memory retrieval paper examines an area that looks to me like adjacent secondary motor cortex rather than ACC proper. Also, the emphasis on "regret" and "regretful situations" really isn't supported by any rigorous evidence. Just because the effect occurs on free choice trials when they choose the on average less preferred option does not automatically confer that the rats must be regretting the choice, particularly as the effect is still present on forced choice trials where there was no other option.What is particularly frustrating about this is that I don't see any reason to try to oversell the finding by surrounding it in this vague and overblown terminology. The basic effect – of representing a potential alternative outcome at the time of reward collection – is very neat and fits well with the wider literature of this region in encoding the value of doing alternative courses of action and updating internal models of the environment.

We appreciate these comments and have edited the manuscript extensively to make it more sober. The focus of the introduction is now shifted away from default mode networks and regret, to the formation of schema (models), representation of space, and control of actions. We address a few of the specific comments through the following:

1) Regret: this term has been removed from the manuscript except for one reference to literature.

2) Default mode network: there is no longer any mention of this in the Introduction. We agree that present data are not sufficient to make any specific claims about the DMN. We do, however, point out in the discussion that the teleportation phenomenon reported here occurs at a time (post-reward) in which replay of events has been reported in other structures (hippocampus, OFC). We do observe that the timing is consistent with that of the default mode network, and speculate that the dynamics of the teleportation/replay events could be a simplified version of that in humans supporting cognition. This is clearly stated as speculation, and therefore feel it is appropriate near the end of the paper. The reason we do so is because the present evidence showing intermixing of a spatial map with other information that could be considered part of the cognitive map. We are unaware of other evidence of such mixing with similar resolution, and this part of the discussion could help readers in other fields relate to the findings.

3) Memory retrieval: We argue that evidence does suggest that the ACC is involved in memory retrieval and the formation of world models (schema). The Rajasethupathy et al., 2015 study shows fluoresence in the dorsal medial PFC reasonably close to where we have recorded. We have now added descriptions and citations to two other papers regarding the role of ACC in the control of memory and formaiton of shema: (1) Ito HT, Zhang et al., (2015); and (2) Wang, Tseand Morris, (2012).

[Editors' note: further revisions were requested prior to acceptance, as described below.]

Reviewer #1:I think the revised manuscript is more tightly focused and improved. The findings are potentially very interesting.While I understand that the authors would have liked to make a stronger connection between behaviour and physiology but maybe cannot given the task design, I would nonetheless like just a bit more data just to show that the animals are performing the task at all as you'd expect. For instance, it's stated that the rats chose one option on ~88% of free choice trials, but it's not as far as I could see stated what these choices were. This is important given the high reward side switches at block 4 in Figure 1B and also there are some blocks where it is straightforward to determine what to do (HR/LE v LR/LE or LR/HE v HR/HE) and others where the contingencies are mixed. Could a summary of the behavioural performance in the 2 rats / 7 sessions analysed for teleportation events be included as part of Figure 1 perhaps?

The relationships among the effort, reward, and choice (as well as neural activity) are complex, and so we are preparing another manuscript to describe them in detail. The present manuscript focuses on the subset of the data that has a sufficient number of simultaneously recorded units needed to study the spatial encoding and dynamics at reward sites. The other manuscript, which is in preparation, deals with economic choice involving effort and reward, and their neural correlates in individual cells. We feel that the behavioral effects of effort and reward belong in that manuscript rather than this one. However, to address your question, we have included Author response image 2 showing the influence of reward and effort on the choice of the high reward arm. The first four bars show the performance when the high reward was on one side and the last two show performance when the high reward was on the other side. The x-axis labels show the ramp height choice offered (i.e., low-high means the ramp was low for the low reward arm and high for the high reward arm). Choices are 80-90% towards the high-reward arm but drop to 40-60% when the ramp is at its highest position. Hence, both reward and effort influence choice, and require multiple analyses and plots to fully convey their relationship.

**Author response image 2. respfig2:** 

I also still was a bit confused by what is now Figure 4A. If the data are the averages from each of the 7 sessions, what is "the feeder" (as in "preference for the feeder in free-choice trials" on the x-axis in 4A)? And why, if the data are averages, are there some sessions with very strong/weak preferences for "the feeder" (i.e., < 25% or > 75%) when the animals always had a strong preference in a block? Does that mean in some sessions animals were seldom choosing the other feeder at all in spite of the difference in values?

Thank you for pointing this out – we apologize for the unnecessarily obtuse description. These are session averages for the choice feeder on the top right side of the track. We changed the text in the figure caption and the plot axes to clarify this. The data were counterbalanced for right/left selection for the spatial decoding needed for excursion detection (these are forced-selection trials), and all of the reward-effort utility combinations were present at each target feeder (but not exactly counterbalanced due to the limited number of trials available in each session). The preference measure comes from the free-selection trials. The rats sometimes had strong choice bias for the right-side feeder, and sometimes for the left.

Reviewer #3:The authors did not adequately address my concerns from the last round. The main point is that the two feeder locations are different in that they have different x-y values but similar in that they both deliver reward and the neurons encode both sources of information. That means that if you set up an analysis to search for differences (e.g. LDA), the activity around the feeders will appear to be different. On the other hand, the most common decoding errors ('teleportations') will reflect the similarities. The new controls they provided do not get at this issue.1) In response to my critiques, the authors show a single LDA panel. LDA finds the most discriminating axes and since the authors showed that the networks strongly encode spatial information, LDA presumably separated the 3 feeder locations based on location-related activity. The LDA figure is therefore just another demonstration of what they already showed in Figure 2. The new controls are of little value (see point 3). The authors need to show that the isolated location representations actually shift independent from the contaminating influence of the common reward-related representations. Admittedly, this will be difficult because (in the Introduction) "Our analysis revealed that many cells involved in the excursion also encoded reward value and/or choice information". If single neurons or the network as a whole encodes both spatial and reward information, the dNN will learn this. As a result, a decoding error is not really an error or a teleportation but the dNN indicating that part of the input is reward related and that part of the input is common to the two feeders. The relative degrees to which spatial and reward information is present likely varies from trial to trial and "teleportations" may be the trials where there is a more reward-like input pattern than a location-like input pattern. The authors therefore need to come up with clever ways to parse the spatial and reward signals. Since single neurons encode both reward and location, extracting neurons is not a solution. Instead, the authors could potentially use the LDA for this. Assuming the LDA axes in Figure 3 are oriented to maximally separate activity based on location (please check if this is correct), activity on 'teleportation' trials could be projected onto these axes. The points from the blue feeder trials should then fall into the red cluster and vice versa. Furthermore, the likelihood of finding a 'blue' point in the red cluster on teleportation trials should be much higher than the likelihood of finding a non-feeder point in the red cluster. If the authors do not like this solution, they could implement some other means to first parse out the contribution of reward encoding before using the dNN or to show that just the isolated spatial representations shifts while the rat is stationary at a feeder.

We apologize that neither previous version of the manuscript sufficiently ruled out other possibilities for our finding that the spatial decoder sometimes shifted from the occupied feeder to the remote feeder. The reviewer suggests that the reward encoding could eclipse the spatial encoding at both choice feeders, and if the reward encoding shared similar features, the decoder could sometimes confuse the two. This is certainly a possibility that we did not adequately address previously. We have added to the manuscript a new set of analyses to test this and another possibility (Results section):

The analyses above do not rule out the possibility that the ACC activity occasionally enters a unique ‘latent state’ when the animal is at either choice feeder. For instance, the reward encoding neurons could strongly activate to overshadow the position information in some instances, and this could produce a pattern that emerges at both feeders but is distinct from the normal encoding at the feeders, and thereby confuses the decoder. We sought to test for this by computing the classification accuracy of ACC patterns among the four classes: feeder A during an excursion (A’); feeder B during an excursion (B’); feeder A not during excursion (A); and feeder B not during excursion (B). If the excursions are because of a transition to a common state from both feeders, then the excursion patterns should be highly discriminable from the non-excursion patterns at the same feeder (A’ from A, and B’ from B), but not discriminable from each other (A’ from B’). We found strong evidence for the former, but not the latter. We used the area under the curve (AUC) of the receiver operator characteristic (ROC) to quantify discriminability of samples from pairs of these conditions. An AUC value close to 0.5 indicates that the patterns from two classes are indiscriminable, whereas an AUC value close to 1 indicates that patterns are highly discriminable by the classifier. AUC values between these limits indicate that features of the patterns are sometimes similar and sometimes dissimilar in at least some dimensions. The discrimination of excursions from non-excursions at the same feeder (A’ from A, and B’ from B) was very high (AUC = 0.94). On the other hand, the patterns during the excursions from the two feeders (A’ from B’) were discriminable at a moderate level (mean AUC = 0.75). This suggests that the excursion patterns are not generated from a common state, but that they sometimes have overlapping features.

A schematic summary of the classification results is shown in Figure 3H. The non-excursion patterns (A, B) are highly discriminable (e.g. Figure 3E-G), as are the excursion patterns from the actual position of the animal (A, A’, and B, B’). The excursion patterns partially overlap with each other (A’, B’), and with the pattern from the unselected feeder (A’, B and B’, A). The latter is supported by the results of the spatial decoder. It thus appears that some features of the encoding shift to be more similar to the unselected feeder during excursion events. Because reward and location were confounded in the experimental design, we cannot rule out the possibility that reward encoding contributes to the phenomena. The inability to fully discriminate the excursion patterns from one another could involve some feature of the reward, such as volume, which flips between the choice feeders during the session.

We also note that similar features that appear at each target feeder would be mapped by the decoder network to the midway point between the feeders, because the objective function is based on distance. The midway point minimizes the error. Our data show, however, that the mapping during excursions goes to the alternate feeder rather than the midway point.

We argue that the preponderance of evidence supports our new weaker claim that the pattern of activity shifts during the excursions to often become more similar to the remote feeder than the presently occupied one. This evidence includes not only the classification analysis, but also the output of the position decoder. In particular, we would expect some excursions to begin or terminate at the starting feeder if they were driven by sensory features of reward. We have edited the discussion heavily, and now include a more sober discussion of the new claim in the Discussion section. We have also softened the tone regarding the shifting of encoding by omitting the term ‘teleportation’ and opting for ‘excursion’ form the present state. We discuss the possibility that the state more strongly resembles the remote feeder (than the occupied feeder) in spatial and possibly other dimensions. We now also point out that spontaneous reactivation of neural patterns reported in other studies often involves different numbers of spikes or a compressed time-course as compared to the patterns during the experience. This is an additional confound that may limit discriminability because the filtering and binning of spike data was set based on the experiential data and was not modified to optimize the classification analysis.

Also, as I asked in the last round, please show a complete error map across the entire maze at a spatial resolution like that shown in Figure 2. If the rat is at the feeder on a test trial the distribution of predicted locations should be localized around the other feeder. Again, this is probably because the dNN is picking up on common reward-related activity, but this map would still be useful.

This plot has been added.

2) I still have some trouble with the precision of the spatial encoding, given the limited training data. Since these neurons have low firing rates and are inconsistent, with such limited training data most of the ~5cm locations that were accurately decoded would never be associated with any activity, let alone consistent activity. The precise spatial encoding may instead be an artifact of how the spiking data was pre-processed. In subsection “Decoding ensemble activity” the authors state that they smoothed the data with a kernel having a s.d. of 150ms and then binned at 50ms prior to performing a square root transform. This would smear out the temporal and spatial inconsistencies in the raw data. By eliminating the inherent trial-to-trial and bin-to-bin inconsistencies and filling-in sections of the maze where no activity was present, the pre-processing may have essentially created a highly continuous and repeatable activity mosaic across the maze. Please repeat the spatial decoding analyses using the raw binned data.

We interpret this concern as: the ACC is encoding only a few key positions on the task (turn here, reward there), but we are smearing these highly position-specific signals over the intervening space in the pre-processing, and so the decoding is artificially accurate because of interpolation from this smeared signal that is not present in action potentials. This is an excellent alternative hypothesis, but we think several lines of evidence indicate this is not the case. First, Fujisawa et al., (2008) show units in mPFC with position-dependent activity fields that individually span only a small fraction of the track. The centers of the spatial fields are distributed over the entire track. This suggests that there should not be any’ dark’ regions of the track that cannot be decoded directly from spikes without smoothing. We use smoothing to make up for our sampling deficit with respect to the entire ACC population. We further argue that the smoothing does not invalidate the present results for four interrelated reasons:

First: we show in Figure 1—figure supplement 1B the effect of kernel width on RMSE. The decoding RMSE was about 15cm with a 50ms width, indicating that the decoding does not fail as the kernel shrinks to near the width of the time bins used. Further, we were able to achieve similar error (as with a 150ms kernel std) with a kernel of 75ms and a different network architecture (not shown).

Second: the width of the kernel (150 ms) is 3.3% of the mean time the rat spends locomoting on the track in each lap (4.5 s). The smoothing is therefore filling only small sampling gaps with respect to the scale of the apparatus. In other words, the filter size is small with respect to the variation of activity patterns over the maze. This is corroborated by the above-mentioned plot Figure 1—figure supplement 1B. The error decreases as the size is increased to a point, and then the error begins to increase at abut 700-1000ms width. This appears to be the point at which the filter size begins to exceed the scale of pattern variation.

Third: the smoothing is performed between time bins, and not between trials. It therefore reduces bin-to-bin inconsistencies but has little effect on trial-to-trial inconsistencies. Therefore, bins beyond several standard deviations of the kernel are effectively independent, as are the same bin from trial-to-trial.

Fourth: the error was significantly increased by jittering the spike times on the order of 25 ms. This was spread out in time by the smoothing (address in more detail below), but nonetheless degraded performance. This suggests that small changes in spike timing on all parts of the track matter, which argues against the proposal that the high-resolution decoding accuracy arises from blending together unique patterns between distant (course-resolution) landmarks.

In sum, the scale of smoothing is small with respect to the variation of patterns over the apparatus (within trial), and there is no filtering across trials. Further, the decoding works well even with much smaller kernel widths. Lastly, the wide distribution of position-sensitive cells across the track in Fujisawa et al., (2008) strongly supports the notion that all track positions are encoded by some cells in ACC. We conclude that we are likely not introducing information that is not already present in the ACC population to an extent that invalidates the results.

A final point is that your question raises several deep questions for which we do not think there are precise answers. What is the appropriate granularity of time for relevance to biological neurons, and what is the nature of temporal filtering performed by neurons? The time binning operation performs a filtering operation, so our choice of bin size affects the results even if we do not use a smoothing kernel. We could conceivably create a more biologically-relevant decoder by using a short causal filter mimicking the integration of post-synaptic potentials and a recurrent network architecture, but the performance again depends on parameterization. For these reasons and the evidence that the results are not an artifact of smoothing, we don’t think that pursuing decoding with a different (nor no) smoothing kernel would benefit the present manuscript.

3) I do not understand the 'similarity' controls. The noise procedures are of limited value since any information on this timescale is completely occluded by the pre-processing. Second, it begs the question of how much noise is useful to make the point? Too much and all decoding would break down, too little and the dNN would be robust to it. Again, the critical control is to extract just the spatial representation components and show that they shift back and forth when the rat is at one feeder.

What constitutes ‘small’ is certainly up for debate. We agree that too much or too little provides no information, and advocate for an empirical answer. We argue that because the amount we used degraded performance of the decoder significantly, that it is at least ‘in the ballpark’ of magnitude that is present in the data most of the time. It is true that much of the effect of the added noise will be smeared out by the pre-processing step, however the same is true for noise and random changes in the original signal. We wanted to know if these could lead to a significant proportion of large errors, as could explain the excursions.

The conclusion is that either because of the pre-processing steps or the ability of the network to separate useful information from noise, the decoder output is robust against small random changes in the signal. Therefore, large deviations in the output of the network are more likely to be due to a significant, or patterned, change in the signal. Of course, it is possible that this breaks down under higher levels of noise, network architecture, or other parameters. Rather than attempt to explore this large space, we focused on the new analysis described in above.

4) In the Introduction the authors discuss the impact of preferences because of unequal effort-reward utilities. First, I am unclear how the unique blocks (subsection “Data collection”) and forced/free trials impacted the selection of training/test trials. If the reward magnitudes do not match there would be less excursions given the decreased (although still not zero) similarity in the reward representations. Second, Figure 4A would only be valid if the neurons and the dNN had the same exposure to the two feeders for the same reason that more training data improves classification.

All analysis of spatial decoding and classification used balanced data that included the same number of trials to the left and right. For the spatial decoding (Figure 1 and Figure 2), an equal number of training data are selected from the pool of all trials. The analysis in Figure 3 and Figure 4 included data selected from forced-selection trials only. Right/Left was always balanced, but we did not have sufficient data to additionally balance reward size and effort and have sufficient trials for model fitting. We would like to know if the excursion occurrence is promoted by the degree of reward or utility similarity among choices. The limitation is the estimation of internal valuation beyond choice preference. With the present task design, it would be difficult to fit a well-validated choice model. We think this is better addressed in a future experiment using less-frequent contingency shifts and a higher proportion of free-choice trials.

The analysis shown in Figure 4A does include balanced exposure to both feeders. Moreover, the high reward was sometimes at the right feeder, and sometimes the left. We altered the labels and legend to indicate that it shows choice preference and excursion probability at the right-hand feeder.

[Editors' note: further revisions were requested prior to acceptance, as described below.]

1) Soften the language about a lack of similarity between feeder representations. The Abstract highlights the multi-plexing of reward and position representations and Figure 4 shows there are a separable reward and position components during excursions. Yet the Results are written as if to completely disregard this possibility. For example:Subsection “Excursions of spatial encoding from the physical position to a feeder”: These analyses do not provide evidence that the excursions are not due to similarity at the feeders. LDA only finds the most discriminating axes and says nothing about other dimensions in which the feeder representations could be similar.Subsection “Excursions of spatial encoding from the physical position to a feeder”: An AUC of 0.75 does not provide convincing evidence "that the excursion patterns are not generated from a common state" in spite of the classifier "that they sometimes have overlapping features". Also, Figure 3H is not helpful and needs to be replaced with actual data.In these places and throughout the manuscript (e.g. Discussion section) the language about a lack of similar components at the two feeders needs to be softened. It is ok that there are similarities because these similarities cannot completely explain the excursions.

We have edited the manuscript throughout to soften the tone, including the four specific lines mentioned above. We do not argue that some features of feeder activity patterns may be similar, and instead focus on whether some features are truly different. We have omitted the statement about the excursion patterns not coming from a common state based on the 0.75 AUC and have replaced figure panel in question with actual data. The data come from a PCA of activity of a middle layer of the decoder network. We take care to point out which results come from untransformed input patterns, and which come from patterns transformed by the decoding network. The excursion patterns are only partially separated in the input patterns but become highly separated in the network. We conclude that the non-linear transformation by the network is able to separate the excursion patterns, but we cannot identify what features are used to do so. The reward may very well figure into the transform.

2) The discussion of shifts and remapping in the Discussion section is confusing. Isn't the situation the same as in the present study whereby there are changes in the ensemble that vary in magnitude across cells? I am not sure how dimensionality reduction plays into all this. Furthermore, Rich and Shapiro used a correlation matrix of ensemble activity on different trials, which is not really a dimensionality reduction technique. Likewise, Ma states: "Multivariate analyses were always performed in the full multidimensional space, but for the purpose of visualization, N-dimensional population vectors were projected down into a 3D space by means of metric multidimensional scaling." I have no idea what the authors are trying to say in this section.

We apologize for the confusion. The differences in methodologies was not central to the point we wanted to make – that the evidence of radical shifts of spatial encoding in mPFC may reflect a process akin to ‘global remapping’ in the hippocampus, which does not retain features of other maps in the same physical space. The current decoding approach may be helpful to assess if any associations (map features) are retained in ACC when such shifts occur.

We have edited the paragraph to read:

“These investigators proposed that this provides a shift in context so as to facilitate learning or utilizing different sets of associations (e.g. action-outcome). It remains unclear whether theses shifts are due to a global remapping of the entire ensemble or only a subset of task-relevant cells. The decoding algorithms demonstrated here may be useful for determining if schemas (a.k.a mental models or cognitive maps) retain associative information about space or other features across such shifts, or if ACC wipes the slate clean in some conditions.”

3) There should be some discussion about why there are no excursions to the central feeder on non-preferred trials.

We have added a section to discuss this in the Discussion section.

“The emergence of excursions exclusively at the target feeders, and not the center feeder, suggests a role in outcome comparison or future choice. Excursions did not terminate at the center feeder, suggesting they do not encode the subsequent action from the target feeder, which is always a return to the center feeder as enforced by gates on the track. It is possible that the excursions reflect an unexecuted plan to move from the occupied feeder to the other. If this were the case, however, we would expect to occasionally observe excursions when the rat is at the center feeder or other location on the track. A possible alternative is that the excursions reflect a mechanism for shifting strategies. The rodent ACC is involved in shifting responses (Joel et al., 1997; Birrell and Brown, 2000), and appears to sustain information over time (Dalley et al., 2004; Takehara-Nishiuchi and McNaughton, 2008). Although speculative, it is therefore possible that the excursions trigger a memory trace in ACC that promotes a response shift at the next visit to the choice point on the track. In other words, the ACC may have made a decision for the next choice while at the target feeder, which could preclude excursions at the center feeder or other intermediate point. The ACC is only one of several dissociated circuits that influence binary choice (Gruber and McDonald, 2012; Gruber et al., 2015), and is posited to bias competition among these other systems (Murray et al., 2017). Excursions may therefore have a probabilistic influence on future choice rather than fully determining it.”